# Aqueous phase conversion of CO$_2$ into acetic acid over thermally transformed MIL-88B catalyst

Waqar Ahmad[1], Paramita Koley[1], Swarit Dwivedi[1,2], Rajan Lakshman[1], Yun Kyung Shin[2], Adri C. T. van Duin [2], Abhijit Shrotri [3] & Akshat Tanksale [1] ✉

Sustainable production of acetic acid is a high priority due to its high global manufacturing capacity and numerous applications. Currently, it is predominantly synthesized via carbonylation of methanol, in which both the reactants are fossil-derived. Carbon dioxide transformation into acetic acid is highly desirable to achieve net zero carbon emissions, but significant challenges remain to achieve this efficiently. Herein, we report a heterogeneous catalyst, thermally transformed MIL-88B with Fe$^0$ and Fe$_3$O$_4$ dual active sites, for highly selective acetic acid formation via methanol hydrocarboxylation. ReaxFF molecular simulation, and X-ray characterisation results show a thermally transformed MIL-88B catalyst consisting of highly dispersed Fe$^0$/Fe(II)-oxide nanoparticles in a carbonaceous matrix. This efficient catalyst showed a high acetic acid yield (590.1 mmol/g$_{cat}$.L) with 81.7% selectivity at 150 °C in the aqueous phase using LiI as a co-catalyst. Here we present a plausible reaction pathway for acetic acid formation reaction via a formic acid intermediate. No significant difference in acetic acid yield and selectivity were noticed during the catalyst recycling study up to five cycles. This work is scalable and industrially relevant for carbon dioxide utilisation to reduce carbon emissions, especially when green methanol and green hydrogen are readily available in future.

Fixation of overabundant atmospheric carbon dioxide is an urgent and essential research area that may lead to climate change mitigation. In 2019, carbon dioxide concentration reached 414.7 ppm, and it is anticipated that it may reach up to 500 ppm in 2050[1]. Therefore, recycling of captured CO$_2$ into value-added products is desirable to avoid the catastrophic consequences climate change caused by CO$_2$ in the atmosphere[2]. Several routes for carbon dioxide conversion have been investigated, but the thermocatalytic CO$_2$ hydrogenation pathway is extremely promising due to its fast kinetics, high productivity, scalability, and selectivity[3]. Synthesis of chemicals such as methane[4,5], methanol[6], formaldehyde[7,8], dimethyl ether[9], gasoline-range hydrocarbons[10], oxymethylene dimethyl ethers[11,12], methyl formate[13],

formic acid[14,15], and acetic acid[16,17] have been investigated in recent years. A CO$_2$-based chemicals industry has the potential to lower the CO$_2$ concentration in the atmosphere while simultaneously providing revenue for offsetting the capture costs. The production of acetic acid (AA) via CO$_2$ hydrogenation is one such route that has recently received attention from researchers.

Acetic acid is extensively used in several industrial applications, including food, chemicals, pharmaceuticals, textile, cosmetics, and polymers[18]. Moreover, it is consumed in the synthesis of vinyl acetate monomer[19], acetic anhydride[20], and cellulose acetate[21]. AA is also used as a solvent during terephthalic acid manufacturing[22,23]. It is a well-known food preservative and is traditionally named vinegar in the food

[1]Department of Chemical and Biological Engineering, Monash University, Clayton 3800, Australia. [2]Department of Mechanical Engineering, The Pennsylvania State University, University Park, PA, USA. [3]Institute for Catalysis, Hokkaido University, Sapporo 001-0021, Japan. ✉e-mail: akshat.tanksale@monash.edu

industry. Commercially, two major production processes are used for the synthesis of acetic acid – chemical and fermentative[18,24]. Among various chemical routes, the most common industrial AA synthesis method is carbonylation of methanol (MeOH), where AA is produced through different processes such as BASF, Cativa, and Monsanto in the presence of homogeneous Cobalt, Iridium, and Rhodium catalysts, respectively. In the Monsanto process, AA is produced from $CH_3OH$ and fossil fuel-derived CO in the presence of $CH_3I$ and homogeneous rhodium-based catalyst[16,18,25]. The main reaction of acetic acid production from methanol and CO is summarized in Eq. 1.

$$CH_3OH + CO \rightarrow CH_3COOH \qquad (1)$$

Gas phase $CO_2$ hydrogenation to AA, PA (propionic acid), and negligible amounts of $C_{+4}$ acids (butyric acid and valeric acid), and other hydrocarbons ($CH_4$ and $C_2-C_4$) has been investigated using intermetallic Ni−Zn catalysts[26]. Authors highlighted 13.4% $CO_2$ conversion with 58.9% and 18.2% selectivity for AA and PA, respectively, over $N_1Z_3-900$ catalyst at 325 °C, 5400 mL.g$^{-1}$.h$^{-1}$ gas hourly space velocity (GHSV), 30 bar pressure, and $H_2/CO_2$ ratio 0.5[26]. Qian et al. reported AA production via hydrocarboxylation of MeOH with carbon dioxide and hydrogen in 1,3-dimethyl-2-imidazolidinone (DMI) solvent over homogeneous Rh and Ru based homogeneous co-catalysts with a combination of LiI promoter and imidazole ligand. The stability of catalyst was dependent on the imidazole ligand. While the authors also report that imidazole played critical role in inhibiting the reverse water gas shift reaction, but the exact role of imidazole in the reaction mechanism was not clear[16]. The same group also showed AA synthesis via the above-described reaction system in the presence of $Rh_2(CO)_4Cl_2$ homogeneous catalyst, LiCl as a co-catalyst, 4-methyl imidazole ligand and LiI as a promoter[27]. This reaction system is highly complex due to the presence of multiple catalysts, stabilizing ligands and organic solvents. In many cases, the authors report a black precipitate, which is not explained but is likely to be the Ru or Rh catalyst, which demonstrates that the system is not stable in these reaction conditions. However, the authors also demonstrated stable catalytic activity of Rh and Ru based homogeneous catalysts for five cycle during AA synthesis via methanol hydrocarboxylation reaction by using Imidazole ligand and LiI promoter in DMI solvent[16]. Hasan et al. reported low yield of AA (1.58 mmol/L) over NiO-C/Al$_2$O$_3$, heterogeneous catalyst at 130 °C and 35 bar total pressure of $CO_2$ and $H_2$ in 1,4 dioxane solvent after 6 h of reaction. Instead a higher amount of formic acid (FA, 4.08 mmol/L) was generated[28]. Therefore, there is an urgent need to develop a stable and active heterogeneous catalyst based on low cost metals for AA synthesis which can be efficient for industrialisation and scaleup.

He et al. report FA and AA production via hydrothermal $CO_2$ reduction with Fe nanoparticles as stoichiometric reagent in which they are converted into ferrous carbonate[29]. In 2021, Wang et al. described the acetate production from direct hydrothermal $CO_2$ hydrogenation in the vicinity of hexagonal closed packed cobalt (HCP-Co) catalyst and NaOH additive. The presence of CoO/Co interface was responsible for $CO_2$ activation followed by C−C coupling. A maximum of 9.5% acetate yield was achieved with HCP-Co after 6 h at 300 °C, 0.5 M NaOH, 40% water filling, and 40 mmol Co under 15 bar $CO_2$ and 35 bar $H_2$ atmosphere (optimized reaction conditions). The reaction pathway was dominated by $^*CH_2$ and HCOO intermediates during $CH_3COO^-$ formation via carbene reaction. Whereas, $^*CH_2$ intermediate appeared from CO hydrogenation reaction[30]. Recently, Yatabe et al. reported a plausible reaction mechanism of AA production through aqueous phase $CO_2$ hydrogenation with $CH_3I$ additive over Rh-based homogeneous water-soluble catalyst, where, it plays the key role as electron storage catalyst. The water solvent is not only used as a green solvent but it also behaves as Lewis base by extracting $H^+$ from hydrogen. In this study, AA turnover number (TON) was very low

(TON = 1) using Rh-based homogeneous catalyst in the presence of $CH_3I$, LiBr (Lewis acid) and mixture of $H_2O/CH_3OH$ (1/1) solvent after 24 h reaction at pH 2.0, and 80 °C under 1.5 bar $CO_2$ and 8 bar $H_2$ atmosphere[31]. To the best of our knowledge, Fe-based heterogeneous catalysts have not been reported for $CO_2$ conversion in aqueous phase. Heterogeneous catalysts have advantages in scale-up, and compares favourably against homogeneous catalysts which require large downstream separation processes.

Here we present a Fe-based thermally transformed metal organic framework catalyst (MIL-88B) for hydrocarboxylation of MeOH to produced AA. Recently, metal organic framework (MOFs) derived carbonaceous materials have been reported for their remarkable catalytic properties[32–34]. Thermal transformation of MOFs results in a carbonaceous material with embedded metal or metal-oxide nanoparticles[34]. As these particles are embedded in the matrix of decomposed organic linkers, they show greater resistance to sintering at higher temperatures. Depending on the thermal treatment, the thermally transformed MOFs have features such as high surface area, porosity, and fine dispersion of metal nanoparticles that are desired in an ideal heterogeneous catalyst. Moreover, the porous carbon framework provides better mass transfer to enhance the reaction rate. In this work, thermally transformed MIL-88B, called T-MIL-88B, consisted of dual active sites−$Fe^0$ and $Fe_3O_4$, accelerating the conversion of $CO_2$ into AA, compared with other Fe-based catalysts tested which contained only $Fe_3O_4$ or $Fe^0$ and $Fe_2O_3$. In this process, AA is produced in a series of reactions[15,16,25] (Eqs. 2−4)−

$$CO_{2(aq)} + H_{2(aq)} \overset{Fe}{\leftrightarrow} HCOOH \qquad (2)$$

$$CH_3OH_{(l)} + LiI \rightarrow CH_3I + LiOH \qquad (3)$$

$$CH_3I + HCOOH + LiOH \rightarrow CH_3COOH + LiI + H_2O \qquad (4)$$

Overall Reaction

$$CO_{2(aq)} + H_{2(aq)} + CH_3OH_{(l)} \xrightarrow{Fe,LiI,150°C} H_3CCOOH + H_2O \qquad (5)$$

## Results and discussion
### Catalyst characterisation
Figure 1a–c illustrates the PXRD diffractograms of the catalysts, before and after catalytic tests. Calcined Fe/CBEA catalyst showed characteristic peaks of α-$Fe_2O_3$, most of which were not observed in the reduced catalyst. Instead, the reduced catalyst showed $Fe^0$ peaks at 2θ = 44.7° and 65° and residual α-$Fe_2O_3$ peaks at 35.98° and 62.83°. However, there were no $Fe^0$ or α-$Fe_2O_3$ peaks detected in the used catalyst which indicated leaching of Fe from the catalyst support. The residual reaction solution slowly turned to red color over a period of few days, indicating presence of iron oxides in the solution. Therefore, Fe/CBEA catalyst was not considered further.

Both the fresh and the used T-Fe/MIL-101 catalyst showed peaks corresponding to $Fe_3O_4$, suggesting that the catalyst was stable after the reaction. However, the α-$Fe_2O_3$ peaks observed in Fe/MIL-101 (Fig. S1a, ESI) which did not reduce to $Fe^0$ in T-Fe/MIL-101.

Figure S1b shows the PXRD pattern of MIL-88B which has good resemblance with literature[35,36]. Characteristic peaks of T-MIL-88B-495, T-MIL-88B-500 and T-MIL-88B-505 catalysts revealed the transition in iron phases after thermal transformation of MIL-88B at different temperatures (Fig. S1c). Diffraction peaks of T-MIL-88B-495 show the presence of mainly $Fe_3O_4$ phases (Fig. S1c), whereas, T-MIL-88B-500 catalyst peaks showed both $Fe_3O_4$ and $Fe^0$ (Fig. 2c and Fig. S1c), while, T-MIL-88B-505 presented major peaks of $Fe_3C$ and $Fe^0$, and few low-intensity peaks of $Fe_3O_4$ phases (Fig. S1c). These results suggested the

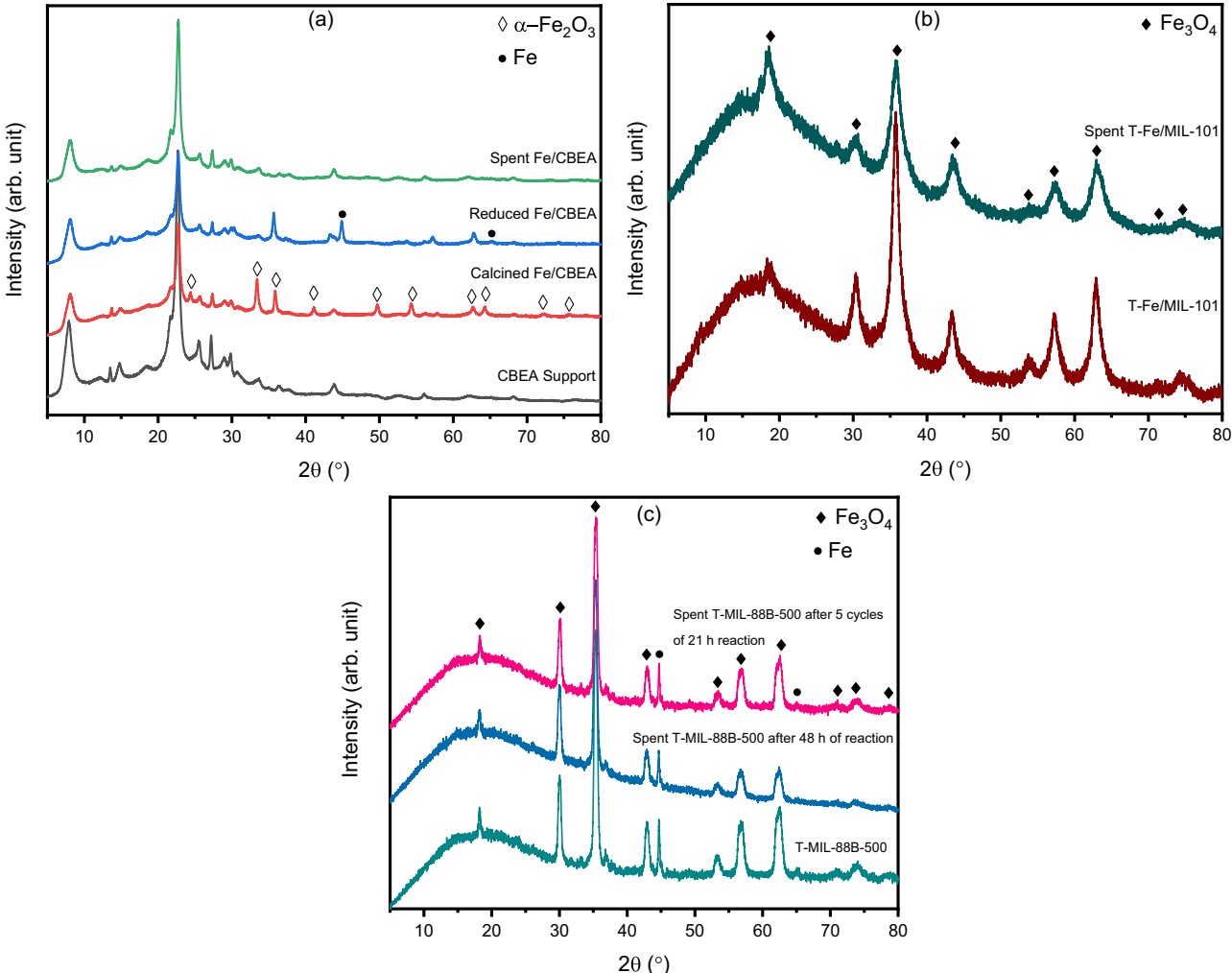

**Fig. 1 | Powder X-Ray Diffraction (PXRD) patterns of the as synthesized and used catalysts used in this study. a** as prepared, calcined, reduced and used Fe/CBEA, **b** as prepared and used T-Fe/MIL-101, and **c** as prepared and used T-MIL-88B catalysts after 48 h reaction in presence of $CH_3I$ and 5 cycles of 21 h each in presence of $CH_3OH$ and LiI.

vital role of temperature during thermal transformation of MIL-88B under reducing atmosphere because a small change in temperature alters the structure of catalyst. Both $Fe_3O_4$ and $Fe^0$ peaks of the T-MIL-88B-500 catalyst remained unchanged after a single run of 48 h reaction time and 5 cycles of 21 h each (Fig. 1c). Only $Fe_3O_4$ peaks have been reported after the thermal treatment of MIL-88B at 500 °C under nitrogen atmosphere[37]. However, due to the reducing atmosphere used in this study, some of iron oxide nanoparticles reduced to $Fe^0$ at 500 °C. No evidence of iron carbide was found in the PXRD results of T-MIL-88B-500.

During the thermal transformation of MOFs, first, the linkers break from the metal oxide clusters. After that, the metal oxide clusters agglomerate and reduce depending upon the chemical environment. Figure 2a shows the MIL-88B structure consisting of the $Fe_3O$ clusters coordinated by six carboxylate ligands and three adsorbed water molecules. The ligands can be water, hydroxylate, or fluorine depending on the synthesis method. Upon heating the MOF, we observe an increase in the density, characteristic of the negative thermal expansion coefficient of the MOF (Fig. 2b). The behaviour is similar to our earlier study of thermal transformations in Zr-based MOFs[38]. At approximately 500 °C, the linkers start detaching from the cluster, and the framework begins to collapse. Consequently, we observe a significant increase in the density of the system. After detachment of the organic linkers, the linkers go through thermolysis,

forming small gaseous molecules such as $H_2$, $H_2O$, CO, and $CO_2$. These molecules were periodically removed from the simulation box to mimic the gas and solid phase separation. The thermally transformed MOF cooled at 300 K is shown in Fig. 2c–e). The MOF treated at 1500 K for 200 ps shows a larger number of oxygenated molecules, mostly present as the carboxylate groups attached to the Fe atoms. However, upon treating the MOF at 2000 K for 500 ps, most of the carboxylate oxygen atoms are removed in the form of CO, $CO_2$, and $H_2O$, and a Fe-$C_x$ matrix is dominant. On the other hand, upon treating the MOF at 1500 K for 500 ps, we observe the carboxylate oxygens partly removed in the form of small gasses, whereas the remaining oxygen atoms are present in the Fe-matrix.

We can better understand the local coordination of Fe atoms by plotting the radial pair distribution function (g(r)). Figure 3a shows the g(r) for the Fe-Fe pair. The characteristic peak of MOF (~3.8 Å) is diminished in all the thermally transformed MOFs, which indicates that the $Fe_3O$ cluster arrangement is broken. The Fe-Fe peaks in the transformed MOFs are similar to the BCC/FCC phase of Fe. The two peaks between 2 Å and 3 Å in the Fe-BCC/FCC structure are merged into one peak in the thermally transformed MOFs, commonly observed in high-temperature quenching of iron[39]. Similar behaviour is observed for the peaks between 4 Å and 6 Å. Figure 4b shows the g(r) for the Fe-O pair. All the thermally transformed MOFs have the highest peak at ~1.5 Å representing the O/OH bonded to Fe atom. Although the

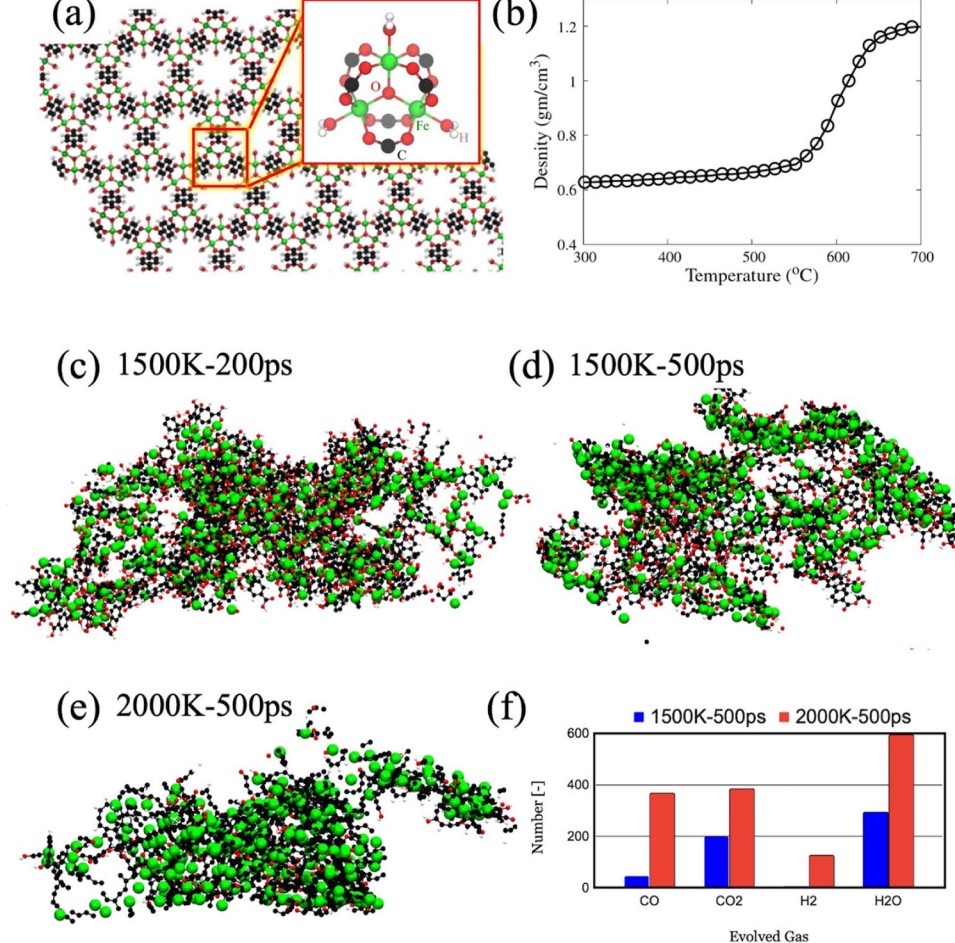

**Fig. 2 | Thermal transformation of MIL-88B(Fe) structure. a** Shows the MIL-88B (Fe) MOF and metal cluster topology in the inset and **b** shows the density change during heating. The final transformed structures cooled at 300 K after the thermal treatment at **c** 1500 K for 200 ps, **d** 1500 K for 500 ps, and **e** 2000 K for 500 ps. **f** The total number of gas molecules removed evolved during the thermal treatment.

MOF treated at 2000 K has the largest peak, this is due to nearly all the remaining oxygen atoms being bonded to Fe atoms in form of O/OH. A peak around ~1.9 Å is observed, prominently for MOF transformed at 1500 K for 500 ps. This peak is similar to oxygen atoms present within the Fe-matrix, similar to the first peak of $Fe_2O_3$ and $Fe_3O_4$. This peak is absent for the other two thermally transformed structures. Based on these results, we conclude that upon heating, first, the carboxylate oxygen is removed from the system, and $Fe_3O$ clusters partly arrange in the form of $Fe_2O_3/Fe_3O_4$ clusters in a Fe and C matrix. However, the interstitial oxygen is removed upon further heating, and a Fe-nanoparticle/Fe-C matrix forms at 2000 K. Figure 3c shows the g(r) for the Fe-C pair. A peak of Fe-C is observed at ~2.4 Å; however, it does not match the Fe-C peak in $Fe_3C$ at ~2 Å. Therefore, carbon is present in the iron matrix, not in the $Fe_3C$ phase.

Figures 4a–f show the TEM images of MIL-101, Fe/MIL-101, T-Fe/ MIL-101, MIL-88B, T-MIL-88B-500, and used T-MIL-88B-500, respectively. MIL-101 shows the characteristic octahedral shape of ca. 200- 300 nm size (Fig. 3a and Fig. S6a, ESI)[40]. After impregnation of Fe over MIL-101, agglomerates of Fe nanoparticles were observed on MIL-101 (Fe/MIL-101) with approximately 50–100 nm in size (Fig. 4b), whereas after thermal transformation, T-Fe/MIL-101 exhibited approximately 5– 30 nm particles (Fig. 4c). The emergence of these smaller nanoparticles is likely due to the thermal transformation of Fe/MIL-101 in reducing atmosphere, where the deconstruction of linkers leads to breakage of the Fe agglomerates. Figure 4d and Fig. S6b (ESI) show the characteristic fusiform rod-shaped morphology of MIL-88B with

~360 nm length and 90 nm width[37]. After thermal transformation, T-MIL-88B-500 shows a narrow range of $Fe^0/Fe_3O_4$ nanoparticle which are well-dispersed over the carbonaceous support (Fig. 4e). The amount of Fe on T-MIL-88B-500 is 49.3%, with 13.7% C and negligible amount of H, N and S (Table S1, ESI), which indicates that original MOF structure is completely transformed into porous carbon. Moreover, CO pulse chemisorption results reveal 0.017% Fe metal dispersion for T-MIL-88B-500 with the metal surface area of 0.055 m²/g. Figure 4f shows that the T-MIL-88B-500 catalyst retains its structure after 48 h of reaction. Figure 4g, h illustrates the particle size distribution (PSD) for T-MIL-88B-500 and used T-MIL-88B-500, respectively. 525 and 476 particles were measured from multiple images which showed most of the particles in 4–16 nm for both fresh and used T-MIL-88B-500, respectively. The peaks were observed at 8 nm with average particle sizes of 9.7 and 9.1 nm for fresh and used T-MIL-88B-500, respectively which suggested that T-MIL-88B-500 is stable and potentially reusable for this reaction.

The surface oxidation state of Fe in the different catalysts was evaluated by X-ray photoelectron spectroscopy (XPS), as shown in Fig. S2 (ESI). For T-MIL-88B-500 (Fig. S2a), Fe $2p_{3/2}$ XPS spectrum exhibited three peaks, including a peak at 706.7 eV corresponding to metallic iron[41]. Moreover, the other two peaks at 710.0 and 712.09 eV which are correlated to $Fe^{+2}$ and $Fe^{+3}$ oxidation state of iron and a satellite peak appeared at 719.07 eV[42]. In the Fe $2p$ region of T-MIL-88B-500, Fe $2p_{1/2}$ and Fe $2p_{3/2}$ peaks at 710.0 and 723.7 eV, respectively, having a spin-orbital splitting of 13.7 eV indicate the presence of $Fe_3O_4$

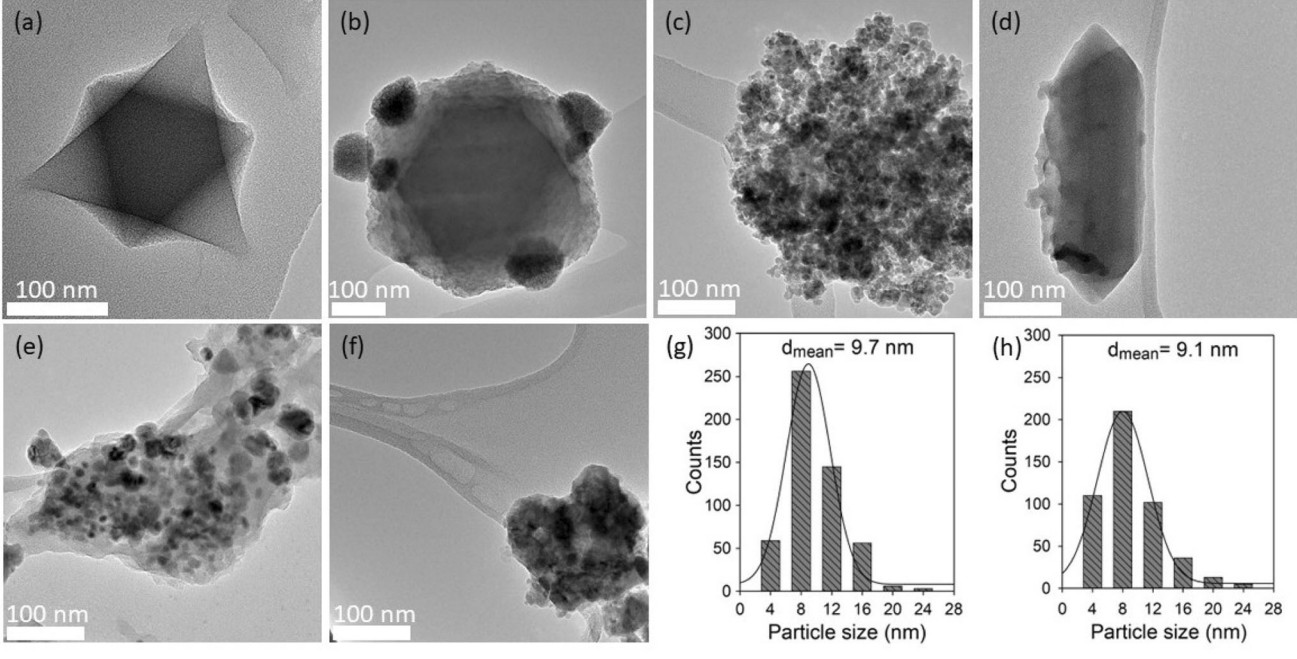

**Fig. 3 | Radial pair distribution function of thermally transformed and parent MIL-88B metal organic framework (MOF). a** Fe-Fe, **b** Fe-O, and **c** Fe-C pairs. The top graph shows the g(r) for the reference materials whereas the bottom graph shows the g(r) for MOF and the thermally transformed MOFs.

**Fig. 4 | TEM micrographs of various catalysts used in this study. a** MIL-101, **b** Fe/ MIL-101, **c** T-Fe/MIL-101, **d** MIL-88B, **e** T-MIL-88B-500, and **f** used T-MIL-88B-500 after 48 h of aqueous phase $CO_2$ hydrogenation reaction in the vicinity of $CH_3OH$ and LiI additives; and particle size distribution of **g** T-MIL-88B-500, and **h** used T-MIL-88B-500.

in T-MIL-88B-500[43]. $Fe_3O_4$ may exist as mixed FeO and $Fe_2O_3$ states, which appears from $Fe^{+2}$ and $Fe^{+3}$ oxidation states[44]. The present XPS study shows that $Fe_3O_4$ is the dominant species on the surface, where the amount of $Fe^{+2}$ was 48.4% and $Fe^{+3}$ was 34.5%, whereas $Fe^0$ was 17.1%. Therefore, the ratio of $Fe^0$ to $Fe_3O_4$ was accounted as 1:4.85 in T-MIL-88B-500.

The XPS spectra of Fe $2p_{3/2}$ in T-Fe/MIL-101 exhibited two peaks at 710.7 and 712.4 eV which is relate to $Fe^{+2}$ and $Fe^{+3}$ along with satellite peak at 719.03 eV. Furthermore, Fe $2p_{1/2}$ and Fe $2p_{3/2}$ of $Fe^{+2}$ appeared at 710.7 and 724.4 eV and the spin-orbital splitting is 13.7 eV which is interpreted as $Fe_3O_4$ in T-Fe/MIL-101. Metallic Fe peak is absent in this catalyst which is in good agreement with PXRD results. For Fe/MIL-101 catalyst, Fe $2p_{3/2}$ XPS spectra also contained both $Fe^{+2}$ and $Fe^{+3}$ at 710.6 and 712.3 eV, respectively. However, the spin–orbit splitting for Fe $2p_{1/2}$ and Fe $2p_{3/2}$ is 14.0 eV (724.6 and 726.5 eV) which suggested the absence of $Fe_3O_4$ phase. For MIL-88B, Fe $2p_{3/2}$ XPS spectra is included both $Fe^{+2}$ and $Fe^{+3}$ peaks at 710.1 and 711.6 eV. Although, the spin–orbit coupling for Fe $2p_{1/2}$ and Fe $2p_{3/2}$ is 13.36 eV (723.57 and 725.3 eV) which confirmed that $Fe_3O_4$ is not present in MIL-88B. Figure S2b represent the Cr XPS spectra of MIL-101, Fe/MIL-101 and T-Fe/MIL-101 catalysts. In MIL-101, Cr $2p$ XPS spectra contained only one peak at 577.6 eV which corresponds to $Cr^{+3}$ oxidation state[45]. For Fe/MIL-101, Cr XPS spectra attributed to two peaks at 577.2 and 578.8 eV which mainly resembles with $Cr^{+3}$ and $CrO_3$[46]. The negative binding energy shift (0.4 eV) of $Cr^{+3}$ as compared to $Cr^{+3}$ present in MIL-101 is most likely due to the interfacial electronic interaction (charge transfer) between Cr and Fe after the inclusion of Fe in MIL-101[42]. The Cr spectra for T-Fe/MIL-101 mainly consisted $Cr^{+3}$ peak at 577.1 eV and the amount of $CrO_3$ is very low as compared to Fe/MIL-101 which may be due to the thermal transformation of Fe/MIL-101 under hydrogen atmosphere that reduces the Cr oxides species on catalyst surface.

The C $1s$ XPS spectra for MIL-88B and Fe/MIL-101 (Fig. S2c) show three different types of C peak at 284.8, 285.6 and 289.0 eV which belongs to C–C, C–O–C and O–C=O, respectively[47]. In the C $1s$ XPS spectra of T-Fe/MIL-101, the O–C=O peak intensity decreased as compared to MIL-88B and Fe/MIL-101. Moreover, the C $1s$ XPS spectra of T-MIL-88B-500 contains only two peaks corresponding to C–C and C–O–C, whereas, the O–C=O peak is absent, which is likely due to the evolution of CO and $CO_2$ during thermal transformation of both Fe/MIL-101 and MIL-88B, reducing the oxygen content in the catalyst.

Figure S3 (ESI) illustrates the XANES analysis of Fe $L_{2,3}$-edge in MIL-88B and thermally transformed MIL-88B at different temperatures and compared with reference $Fe_2O_3$. MIL-88B showed one major peak at 706 eV, with a shoulder at 708 eV for Fe $2p_{3/2}$, where the high intensity 706 eV peak is most likely related with the $Fe_3O$ coordinated iron complex[48]. However, the emergence of shoulder peak highlighted the existence of higher oxidation state of iron in MIL-88B. Here, T-MIL-88B-495, T-MIL-88B-500 and T-MIL-88B-505 exhibited the high-intensity peak at 708 eV which is consistent with $Fe_2O_3$ peak. However, the shoulder at 706 eV observed in thermally transformed MIL-88B may be attributed to partially reduced iron oxide. T-MIL-88B-505 showed the most intense shoulder at 706 eV, which indicates higher degree of reduction of iron particles with increasing temperature during thermal treatment of MIL-88B.

A thermogravimetric analysis of Fe/MIL-101 and MIL-88B under Ar atmosphere is shown in Fig. S4a (ESI). For Fe/MIL-101, the weight loss in the range of 50–250 °C is because of the evaporation of water and removal of free terephthalates inside the pores of MOF[49]. Thereafter, the main weight loss in the temperature range of 270 to 670 °C is due to the degradation of organic ligand in the framework of MOF which is attributed to the collapse of the framework[49]. The weight loss of MIL-88B before 250 °C corresponds to the removal of water and excess DMF from the framework[36]. Further weight loss in the temperature range of 300 to 500 °C is due to the degradation of $H_2BDC$ and the breakdown of the framework. The step in the TGA profile of between

550–650 °C is most likely due to the carbonization of the framework and the formation of $Fe_3O_4$–carbon composites[36].

Figure S4b (ESI) compares TG and DTG analysis of MIL-88B under $5\%H_2/N_2$ and pure $N_2$ atmosphere. No significant difference in weight loss was observed below 400 °C for the two cases. However, the weight loss between 400-495 °C is higher and faster under the reducing atmosphere. In the $5\%H_2/N_2$ atmosphere, the higher intensity DTG peak at 450 °C is believed to be due to the reduction of $Fe_3O_4$–carbon composites, which is in good agreement with the PXRD of T-MIL-88B-495 (Fig. S1c) and published PXRD of MIL-88B treated at 500 °C under $N_2$ atmosphere[37]. Moreover, an additional DTG peak emerged at 495–530 °C under $5\%H_2/N_2$ mixture which shows continuing reduction to form $Fe^0$-carbon and $Fe_3C$-carbon composites as seen in PXRD results of T-MIL-88B-500 and T-MIL-88B-505, respectively (Fig. 1c and Fig. S1c). For $N_2$ environment, no derivative weight loss peak was observed between 495-565 °C. However, a sharp peak was detected between 565–650 °C which might be due to the formation of $Fe^0$-carbon composite reported in the literature for MIL-88B treatment at 600 °C and 700 °C under $N_2$ atmosphere[37]. Overall, the weight loss of MIL-88B under $5\%H_2/N_2$ gas mixture atmosphere (84.4%) is higher as compared to $N_2$ environment (79.7%).

Figure S5 (ESI) presents $CO_2$-TPD results of T-MIL-88B-500 catalyst to measure the basicity of the catalyst. The surface basic sites were categorized as weak (50–212 °C), medium (212–328 °C), and strong (328–458 °C) basicity. In addition to the support, iron oxide nanoparticles may also adsorb $CO_2$[50,51]. Overall, total quantity of desorbed carbon dioxide was 16.29 μmol/g from T-MIL-88B-500, where, the amount of $CO_2$ desorbed from weak, medium and strong basic sites were 12.24, 1.92 and 2.13 μmol/g, respectively.

## Role of Fe-based zeolite and MOF catalysts

Figure 5a–c illustrates the yield and selectivity of AA via aqueous phase $CO_2$ reduction with iodomethane at various pressures. All the catalysts showed some activity for AA production; however, T-MIL-88B-500 was clearly the most active and selective catalyst with the best yield of 504 mmol/$g_{cat}$.L and AA selectivity of 92.4%. Based on stoichiometric calculation, it is equivalent to 80.6% conversion of $CH_3I$ into AA. Both Fe/CBEA and T-Fe/MIL-101 provide lower activity for $CO_2$ hydrogenation and >90% selectivity for FA production. With increasing pressure, the yield increased initially but the AA selectivity peaked at 60 bar for both Fe/CBEA and T-Fe/MIL-101. However, the AA yield and selectivity increases with increasing pressure for T-MIL-88B-500. Since Fe was present in the structural framework of T-MIL-88B-500, the thermally transformed catalyst consists of embedded active metal sites dispersed evenly in a carbon matrix[37]. The high AA activity and the selectivity over T-MIL-88B-500 catalyst is most likely due to the presence of both $Fe^0$ and $Fe_3O_4$ which assist the hydrogenation and C–C coupling reactions, respectively[52,53]. Recently, Wang et al. reported the acetate production through formate ($HCOO^-$) and carbene ($^·CH_2$) intermediate reaction pathway using hexagonal closed packed cobalt (HCP-Co) catalyst and NaOH additive which provided maximum 9.5% acetate yield after 6 h of reaction at 300 °C, 0.5 M NaOH, 40% water filling, and 40 mmol Co under 15 bar $CO_2$ and 35 bar $H_2$ atmosphere. However, the catalyst was inactive at lower temperature (<200 °C)[30]. In comparison, this study shows 504 mmol/$g_{cat}$.L AA yield with 92.4% AA selectivity after 21 h of reaction over T-MIL-88B-500 at 150 °C, $H_2$/$CO_2$ = 1, $CH_3I$ = 10 mmol, catalyst amount = 400 mg, $H_2O$ = 40 mL and stirring speed = 200 RPM.

## Extent of reaction with time

Figure 6a illustrates the extent of reaction over T-MIL-88B-500 to produce AA and FA via $CO_2$ hydrogenation with $CH_3I$ as the starting material in the aqueous media. The reaction proceeds via formation of FA as the initial product, whereas AA was not detected until after 8 h of reaction. The AA yield and selectively sharply increased between 12 to

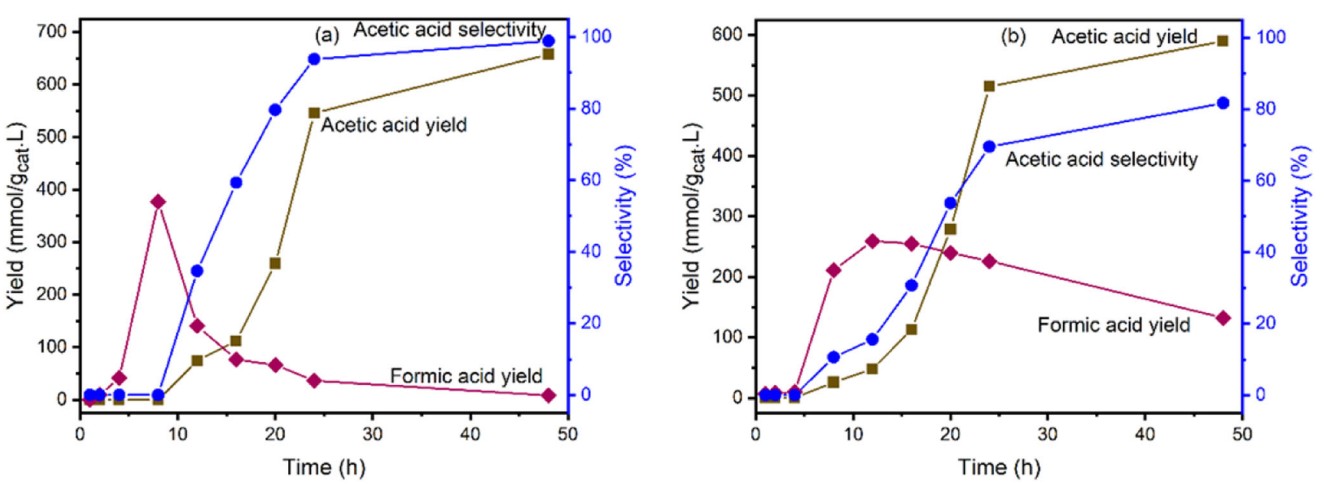

**Fig. 5 | Yield and selectivity of formic and acetic acids produced on various Fe-based catalysts during aqueous phase CO₂ hydrogenation in the presence of CH₃I additive. a** Fe/CBEA, **b** T-Fe/MIL-101, and **c** T-MIL-88B-500. Reaction conditions: $T = 150\,°C$, $H_2/CO_2 = 1$, $t_R = 21\,h$, $CH_3I = 10\,mmol$, amount of Fe/CBEA = 1 g, amount of T-Fe/MIL-101 = 400 mg, amount of T-MIL-88B-500 = 400 mg, $H_2O = 40\,mL$ and stirring speed = 200 RPM.

**Fig. 6 | Effect of reaction time on carboxylic acids yield and selectivity via aqueous phase CO₂ hydrogenation over T-MIL-88B-500 in the presence of various additives. a** CH₃I (10 mmol), and **b** CH₃OH (10 mmol) and LiI (10 mmol).

Reaction conditions: $T = 150\,°C$, $H_2/CO_2 = 1$, catalyst amount = 400 mg, $H_2O = 40\,mL$, $P_{total} = 70\,bar$ at room temperature and stirring speed = 200 RPM.

24 h, thereafter gradually increasing to 657.6 mmol/g$_{cat}$.L and 98.8%, respectively, at 48 h as the reaction approached equilibrium conversion. Based on the initial CH₃I concentration (10 mmol), 100% conversion at 100% selectivity for AA was achieved, within the range of

measurement errors. However, as discussed later, CO₂ first converts into FA and after reaching the maximum yield (377.4 mmol/g$_{cat}$.L) at 8 h, the FA yield decreases sharply until the end of reaction at 48 h when the FA yield was measured at 8.1 mmol/g$_{cat}$.L. However, since

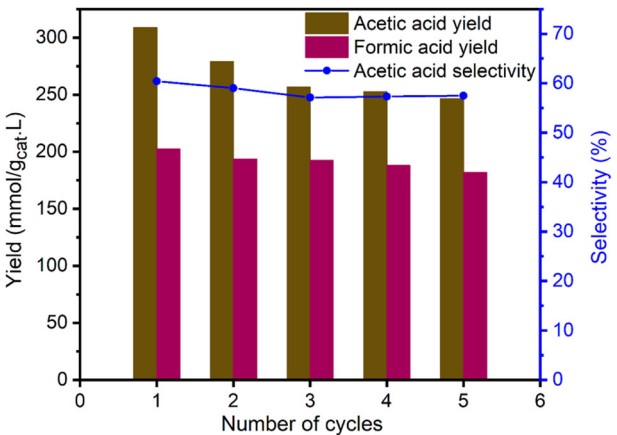

**Fig. 7 | Recycling study of T-MIL-88B-500 via aqueous phase $CO_2$ hydrogenation in the presence of $CH_3OH$ and LiI additives.** Reaction conditions: T = 150 °C, $H_2/CO_2$ = 1, $t_R$ = 21 h, $CH_3OH$ = 10 mmol, LiI=10 mmol, catalyst amount = 400 mg, $H_2O$ = 40 mL, $P_{total}$ = 70 bar at room temperature and stirring speed = 200 RPM.

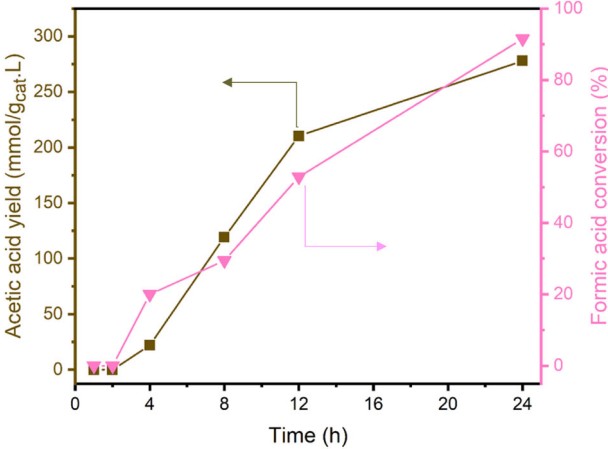

**Fig. 8 | Acetic acid production through HCOOH and $CH_3I$ reaction in water over T-MIL-88B-500 in the presence of hydrogen.** Reaction conditions: T = 150 °C, catalyst amount = 400 mg, $n_{HCOOH}$ = 5 mmol, $n_{CH_3I}$ = 10 mmol, $V_{H_2O}$ = 40 mL, $P_{H_2}$ = 35 bar at room temperature and stirring speed = 200 RPM.

$CH_3I$ is consumed by this time, the residual FA cannot convert into AA. Therefore, for the $CO_2$ hydrogenated into carboxylic acids, the selectivity of AA is 98.8%.

When $CH_3OH$ (10 mmol) was used as a reactant with LiI as a co-catalyst (Fig. 6b), in otherwise identical reaction conditions, the reaction generates in situ $CH_3I$ and hence the peak of FA is broader than Fig. 6a. The AA yield and selectivity increased more gradually and achieved a similar yield of 590.1 mmol/$g_{cat}$.L at 81.7% selectivity after 48 h, which is equivalent to 94% conversion of $CH_3OH$ into AA. The in situ production of $CH_3I$ slowed down the conversion of FA into AA, which may be due to mass transfer limitation.

## Catalyst reusability
Figure 7 shows that the catalytic activity dropped initially but after three cycles, there was no significant decline in AA yield and selectivity. The PXRD of the used catalyst after five cycles (Fig. 1c), and the TEM image (Fig. 4f) and PSD (Fig. 4h) of used catalyst after 48 h confirmed that the structure is stable and there was no sintering or agglomeration of $Fe^0$ and $Fe_3O_4$ nanoparticles in T-MIL-88B-500. The initial loss in activity is likely due to the loss of small particles of the catalyst which could not be recollected in centrifuge.

## Proposed reaction pathway
Reaction mechanism of hydrocarboxylation of methanol in an organic solvent proceeds via reaction of $CH_3OH$ with LiI to produce $CH_3I$ and LiOH which is similar to the carbonylation of methanol (Monsanto process) followed by formation of $CH_3Rh$*I due to the oxidative addition of $CH_3I$ into a Rh* complexing catalyst[16]. Further, $CO_2$ is inserted into $CH_3$-Rh bond to produce $CH_3COORh$*I. Finally, $CH_3COOH$ is formed via reduction of $CH_3COORh$*I with $H_2$ molecule in the presence of Ru* to produce HI as an intermediate. Whereas, LiI is regenerated in situ via HI formation which reacts with LiOH to produce $H_2O$ and LiI. However, here we show aqueous phase methanol hydrocarboxylation in which the reaction pathway deviates from the published works and FA is formed as an intermediate.

First, we show that FA can react with $CH_3I$ in water over T-MIL-88B-500 in $H_2$ atmosphere (Fig. 8). The conversion of FA closely follows AA yield and after 24 h of the reaction FA conversion of 91.5% is achieved with 100% AA selectivity.

Next, we show aqueous phase hydrocarboxylation of $CH_3OH$ using T-MIL-88B-500 as catalyst and LiI as co-catalyst. Here both liquid and gas samples were collected after 48 h of reaction. The liquid sample showed only the presence of HCOOH and $CH_3COOH$ with 81.7% acetic acid selectivity (Fig. 6b). Whereas gas analysis did not detect any

carbonaceous molecules apart from $CO_2$ (ESI, Fig. S8a, b), which eliminates the methanol carbonylation route for AA production.

Figure 9 shows the proposed reaction pathway for acetic acid production via hydrocarboxylation of $CH_3OH$ over T-MIL-88B-500. $CO_2$ and $H_2$ adsorbed over the catalyst and converted into FA, which may desorb. Subsequently, the adsorbed formate species reacts with iodomethane ($CH_3I$) to allow C–C coupling reaction to take place which generates an acetate species and HI as the by-product. Finally, acetate species is converted into acetic acid, whilst LiI might be regenerated from LiOH and HI (step 8). The reaction mechanism of acetate production via formate (HCOO⁻) and ˙$CH_2$ intermediates promotes acetate formation over hexagonal closed packed cobalt (HCP-Co) catalyst during $CO_2$ hydrogenation reaction[30]. Additionally, Yatabe et al. described AA synthesis via $CO_2$ hydrogenation with $CH_3I$ additive in aqueous phase using water-soluble Rh-based homogeneous catalyst via $^{13}CH_3I$ isotopic labelling experiment to confirm the presence of $^{13}CH_3COOH$[31].

In conclusion, we show that thermally transformed Fe-based metal organic framework-based catalyst (T-MIL-88B-500) exhibited high catalytic activity and stability for aqueous phase $CO_2$ transformation into acetic acid. Here, the catalytic activity and the structural property of T-MIL-88B-500 were compared with Fe/CBEA and thermally transformed Fe supported MIL-101 (T-Fe/MIL-101). The T-MIL-88B-500 consisted of both $Fe^0$ and $Fe_3O_4$ phases, which catalyse hydrogenation and C–C coupling reactions, respectively, making this catalyst superior to the others tested here. We present an atomistic mechanism of MIL-88B thermal transformation by ReaxFF molecular dynamics simulations. First, the carboxylic group in the linker breaks to form CO and $CO_2$ forming Fe/FeOx clusters embedded in a carbonaceous matrix. At higher temperatures, metal oxide further reduces to Fe nanoparticles. The presence of sodium and other non-volatile impurities increases substantially in a thermally transformed MOF. The effect of such impurities on catalyst morphology may be explored in future research. In reaction experiments, $CH_3OH$, $CO_2$, and $H_2$ aqueous phase reactants and LiI promoter resulted in a maximum acetic acid yield of 590.1 mmol/$g_{cat}$.L, with 81.7% selectivity after 48 h at 150 °C. We propose that the hydrocarboxylation of methanol to make acetic acid is mediated by the formate route, which is evidenced by formic acid as an intermediate. The T-MIL-88B-500 catalyst was active for at least five cycles for acetic acid production without showing any signs of deactivation via sintering, oxidation, or phase change.

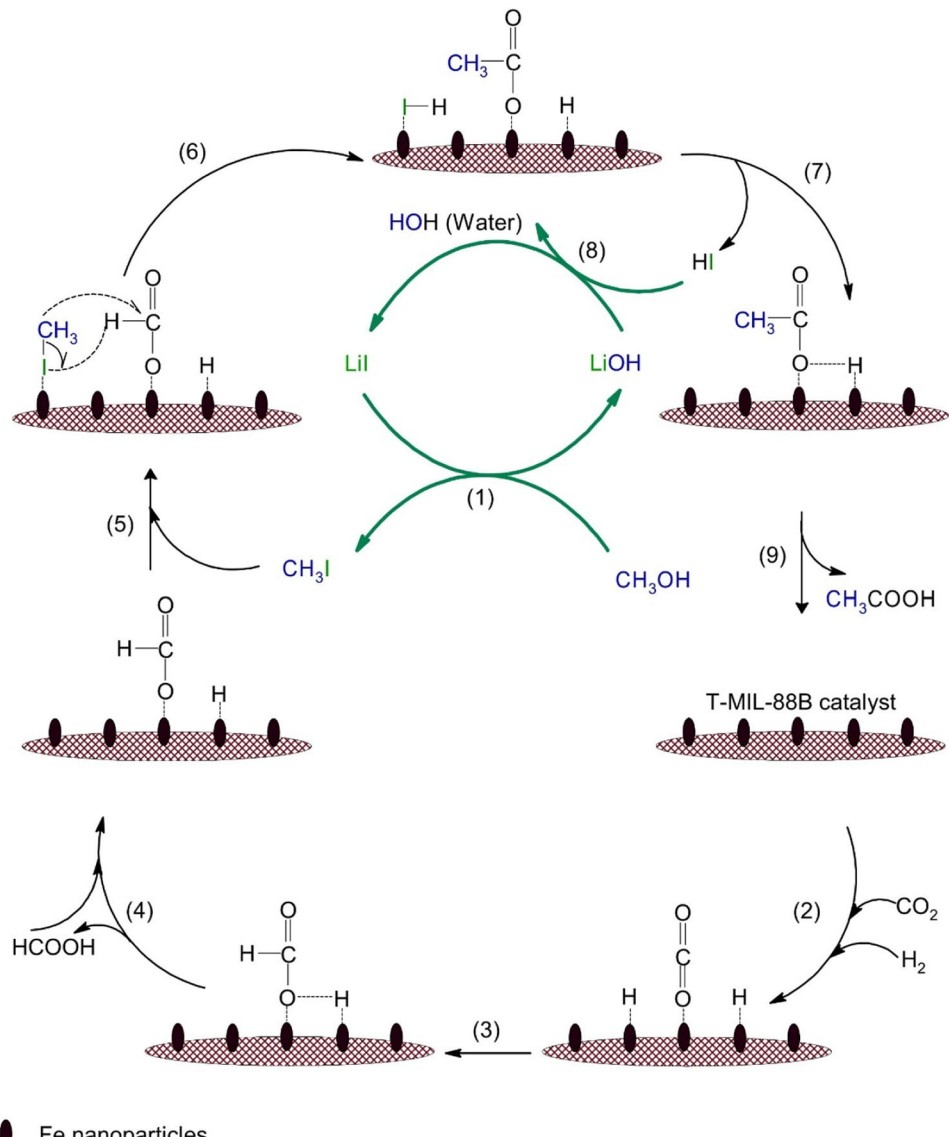

**Fig. 9 | Possible reaction route for acetic acid production via aqueous phase $CO_2$ hydrogenation in the vicinity of methanol and LiI additives over T-MIL-88B-500 catalyst.**

## Methods

### Materials

Iodomethane (CH$_3$I, 99.5%), formic acid (HCOOH, ≥95%), lithium Iodide (LiI, 99.9%), terephthalic acid (H$_2$BDC, 98%), chromium chloride hexahydrate (CrCl$_3$.6H$_2$O, 98%), and iron nitrate nonahydrate (Fe(NO$_3$)$_3$.9H$_2$O, 98%) were purchased from the Sigma Aldrich. Commercial zeolite-beta (CBEA, SiO$_2$/Al$_2$O$_3$ = 38) was received from Zeolyst International. Methanol (HPLC grade) was obtained from the Scharlau Chemicals. Milli-Q water was used for catalysts synthesis (MIL-101 and Fe/CBEA) and acetic acid production experiments.

### Catalysts synthesis

Wet impregnation process was used for Fe/CBEA synthesis as described in our previous publication[11]. The loading of Fe was fixed as 10 wt% in this catalyst. Typically, Fe(NO$_3$)$_3$.9H$_2$O (7.2 g) was dissolved in Milli-Q water (30 mL) by using 100 mL Schott bottle and stirred for 15 min at 65 °C to prepare a homogeneous mixture of Fe solution. Thereafter, 9.0 g of CBEA support was immersed in this solution under stirring and maintained it for 6 h at the same temperature to achieve an even dispersion of Fe particles on CBEA support. The mixture was dried in oven at 100 °C followed by calcination at 550 °C with 5 °C/min for 5 h in

muffle furnace. Vertical tube furnace (50 cm length) and stainless steel (SS) reactor tube (length = 62 cm and outer diameter (OD) = ½ inch) were employed for reduction of catalyst. For this purpose, half of SS tube was firstly filled with quartz wool. Then, approximately 2 g of calcined Fe/CBEA catalyst was added in it and assembled in the SS reactor tube which was fitted to a homemade rig with gas controllers and tube furnace. The catalyst was reduced in the environment of H$_2$/ Ar (1:1 v/v) gas mixture at 400 °C for 5 h with heating rate of 5 °C/min prior to carbon dioxide conversion experiment.

10 mmol of H$_2$BDC and 10 mmol CrCl$_3$.6H$_2$O were poured into a Teflon-lined autoclave. Subsequently, Milli-Q water (72 ml) was added to it. The reaction mixture was sonicated for 30 min followed by stirring for another 30 min at 500 rpm. Thereafter, the autoclave was kept in the oven at 205 °C for 24 h and allowed to cool to room temperature. The resulting solid suspension was transferred into a centrifuge tube. Initially, the centrifugation was performed at 138 relative centrifugal force (RCF) or g force for 3–4 min to remove the unreacted H$_2$BDC present in the reaction mixture. Thereafter, the centrifugation was carried out at 3444 g force for 10 min. The solid sample was then washed with dimethylformamide (DMF) three times and then dried in an oven at 70 °C for 12 h. The synthesised material was named as MIL-101.

For Fe/MIL-101 synthesis, 2.7 g of MIL-101 was suspended in 70 ml ethanol in a Schott bottle and sonicated for 30 min. Separately, 2.17 g Fe(NO$_3$)$_3$.9H$_2$O was dissolved in 20 ml ethanol in a different Schott bottle and stirred for 15 min. The latter solution was poured into the former suspension of MIL-101 in ethanol. Then the Schott bottle which contained Fe(NO$_3$)$_3$ solution was washed with 10 ml ethanol three times and poured into MIL-101 suspension each time to ensure complete transfer of the Fe precursor. The resultant mixture was sonicated for 30 min followed by stirring at 50 °C at 500 rpm for 5–6 h. Finally, the resulting reaction mixture was dried in an oven at 80 °C for 2–3 days. The synthesized catalyst was named as Fe/MIL-101. The Fe loading was fixed as 10 wt% in the synthesized catalyst. Prior to catalytic activity test, almost 2 g of this catalyst was thermally transformed under 100 ml/min H$_2$/Ar (1:1) gas mixture at 500 °C for 5 h with a heating rate of 5 °C/min and allowed to cool in 50 ml/min Ar atmosphere and denoted as T-Fe/MIL-101.

A modified hydrothermal method from literature[37] was adopted for synthesis of MIL-88B. In a typical procedure, 12.12 g of Fe salt (Fe(NO$_3$)$_3$.9H$_2$O) was dissolved in 75 ml DMF under stirring (500 RPM) in a Schott bottle. Separately, H$_2$BDC (4.98 g) and DMF (75 ml) were added in a 250 ml Teflon-liner under stirring (500 RPM). Both Fe and H$_2$BDC solutions were stirred further for 15 min at room temperature. The Fe solution was then poured into H$_2$BDC precursor solution. 12 ml NaOH solution (4.0 M) was slowly transferred into Fe and H$_2$BDC solution mixture and stirred again for 30 min at room temperature. Thereafter, the Teflon-liner was sealed in an autoclave and heated to 100 °C for 24 h. After cooling to room temperature, MIL-88B particles were collected from this mixture via centrifugation at 6750 g force for 10 min and washed three times separately with DMF and methanol, respectively. Finally, the as synthesized MIL-88B was dried overnight in the oven at 80 °C and denoted as MIL-88B. Thermal transformation of MIL-88B was conducted in an identical vertical tube furnace attached with SS reactor tube as described before. 2 g of MIL-88B was thermally transformed at 495 °C, 500 °C or 505 °C for 5 h with a ramp of 5 °C/min under 100 ml/min H$_2$/Ar (1:1 v/v) environment followed by cooling to room temperature under Ar at 50 ml/min atmosphere and denoted as T-MIL-88B-495, T-MIL-88B-500 and T-MIL-88B-505.

## Catalyst characterisation

The crystal structure of the materials was investigated with Powder X-ray diffraction (PXRD) using a Rigaku MiniFlex. Prior to the analysis, Fe/CBEA, Fe/MIL-101, and MIL-88 were reduced or thermally transformed as described above. The powder catalysts were loaded in a zero-background sample holder and scanned between 2–80° 2θ with 4°/min scan speed at 15 mA and 40 kV, except for MIL-88B which was performed at 0.25°/min scan speed. Baseline correction of MIL-88B XRD was performed with OriginPro 2018 software. Nitrogen physisorption analysis was conducted with Micromeritics 3Flex 3500 machine to find the type of adsorption isotherm, Brunauer-Emmett-Teller (BET) surface area and Barrett-Joyner-Halenda (BJH) pore distribution. Tecani T20 was used to capture the transmission electron microscopy (TEM) images of the catalysts. All the samples were dispersed in ethanol and immobilised onto the surface of a holy carbon grid followed by drying in air prior to analysis. ThermoScientific K-Alpha machine was utilized for X-ray photoelectron spectroscopy (XPS) at 1486.6 eV Ephoton and coupled with monochromatic Al Kα radiations. The binding energy (B.E.) baseline correction was conducted by adjusting the C 1s peaks at 284.8 eV. Thermally transformed samples were prepared ex situ prior to the XPS characterization. Fe L$_{2,3}$-edge X-ray absorption near edge structure (XANES) study of MIL-88B and T-MIL-88B samples were conducted in the Advanced Light Source, Lawrence Berkeley National Laboratory, in the Soft X-ray beamline 7.3.1. An energy range from 700 eV to 735 eV was measured at the end-station with a total pressure of 2.1 × 10$^{-8}$ Torr using a 500 mA

ring current with 0.1 eV increments and 5 s count time. The XANES data were collected using total electron yield (TEY) mode, which measures the sample drain current as a result of photo and Auger electrons leaving the sample surface. Shimadzu DTG-60H thermogravimetric analyser was used to check the thermal stability of Fe/MIL-101 and MIL-88B. Both samples were analysed in the temperature range of 100–800 °C with a ramp of 5 °C/min under Ar atmosphere. MIL-88B thermal stability was evaluated in TA SDT 650 thermal analyser. 5 mg of MIL-88B sample was placed in an alumina pan and heated from room temperature to 800 °C with 5 °C/min heating rate under 100 mL/min 5%(v/v)H$_2$/N$_2$ gas mixture and ultra-high purity N$_2$ atmosphere. The surface basic sites of T-MIL-88B-500 were measured with carbon dioxide temperature programmed desorption (CO$_2$-TPD) technique in a Micromeritics AutoChem II. Approximately 500 mg catalyst was preheated at 500 °C with 20 °C/min heating rate and kept at this temperature for 30 min, followed by cooling to 50 °C. Thereafter, 10% (v/v)CO$_2$/He gas mixture was passed through sample at 50 ml/min for 60 min, followed by He (50 ml/min) as a purge gas for 30 min before doing the CO$_2$-TPD analysis between 50–500 °C at 10 °C/min heating rate under He flow. OriginPro 2018 software was chosen for baseline correction. CO pulse chemisorption was conducted (Micromeritics AutoChem II) to estimate Fe dispersion in T-MIL-88B-500 samples. Approximately 500 mg of sample was loaded in the sample tube and pretreated under 10%H$_2$/Ar gas mixture (50 ml/min) for 30 min at 500 °C with 10 °C/min. and purged with He (50 ml/min) for 30 min, followed by cooling to ambient temperature. Subsequently, it was heated to 35 °C at 5 °C/min and maintained at this temperature during the analysis, where, 10%(v/v)CO/He gas mixture (20 ml/min) and He (50 ml/min) gases were utilized as loop and carrier gases, respectively. Pulse gas injection was repeated 10 times at 6 min interval during this analysis.

## Aqueous phase CO$_2$ conversion

All the aqueous phase CO$_2$ conversion experiments were performed in a 100 mL Teflon-lined autoclave batch reactor (Amar Equipment, M4). Typically, 400 mg of thermally transformed catalyst (T-MIL-88B-500) and 40 mL water was added to the reactor and CH$_3$I (10 mmol) was carefully poured into it and sealed. It was purged with H$_2$ three times to eliminate air from the headspace. The reactor was then pressurised with CO$_2$ up to 35 bar, followed by H$_2$ up to a total pressure of 70 bar at room temperature to achieve CO$_2$:H$_2$ ratio of 1:1. The reactor was heated to 150 °C under continuous stirring at 200 RPM for 21 h. After 21 h of reaction, the reactor was allowed to cool to room temperature and the remaining gases were carefully vented from it before dissembling it. The catalyst was recovered from the liquid product mixture by centrifugation at 9953 g force for 1 h. The same procedure was repeated for different total pressures at equimolar CO$_2$:H$_2$ ratio and different catalysts (T-Fe/MIL-101 and Fe/CBEA). Aqueous phase CO$_2$ transformation reaction was also performed at different time intervals employing the best catalyst to check the extent of reaction against time at 150 °C, equimolar CO$_2$:H$_2$ at 70 bar with 200 RPM stirring speed. The liquid samples were analysed using an HPLC (Agilent 1220 Infinity) equipped with a C18 column and a refractive index detector (RID), using 0.5 mM H$_2$SO$_4$ aqueous solution as the mobile phase. The product yields (mmol/g$_{cat}$.L) and selectivity (%) were calculated using Eqs. 6 and 7, respectively.

$$\text{Product}_i\,\text{Yield} = \frac{n_i}{m_{cat} \cdot V_{H_2O}} \tag{6}$$

$$\text{Product selectivity} = \frac{n_i}{\sum_i n_i} \times 100 \tag{7}$$

Where $n_i$ = moles of product, $i$ = HCOOH or CH$_3$COOH, $m_{cat}$ = mass of catalyst (g) and $V_{H_2O}$ = volume of water (L).

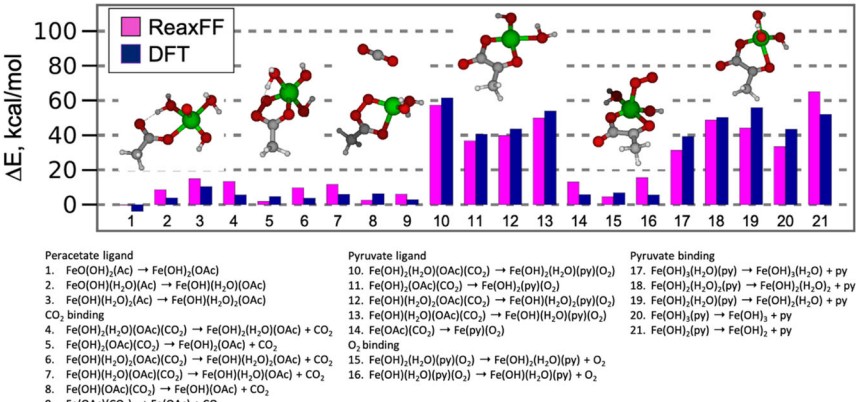

**Fig. 10 | Comparison of ReaxFF and DFT for Fe(II)-complexes containing OH, H₂O, CO₂, O₂, acetate, peracetate and pyruvate ligands.** Ac, OAc and py stand for acetate anion, peracetate anion and pyruvate, respectively. Snapshots are Fe-complexes of acetate, peracetate, CO₂, pyruvate, O₂, and OH ligands, respectively.

The best catalyst was also evaluated for aqueous phase conversion using $CO_2$, $H_2$ and methanol (10 mmol) as reactants and lithium iodide (10 mmol) as the promoter. All other reaction conditions were identical to the previously described procedure.

### Catalyst recycling study
The catalyst recyclability was investigated using $CO_2$, $H_2$ and $CH_3OH$ (10 mmol) as reactants and LiI (10 mmol) as the promoter at 150 °C, equimolar $H_2/CO_2$ with 70 bar pressure at room temperature and 200 RPM stirring speed for 21 h in each cycle. After each cycle, the catalyst was recovered from the product mixture via centrifugation at 9953 $g$ force for 1 h, and without any intermediate treatment, resuspended into a fresh reaction mixture at the same initial conditions. After five cycles, the centrifuged catalyst was dried overnight in oven at 70 °C and stored in an air tight glass vial for characterisation.

### Reaction mechanism investigation
Reaction mechanism was explored by designing two different experiments—(1) using FA and $CH_3I$ as reactants and experiment was conducted in water by using T-MIL-88B-500 catalyst at 150 °C under 35 bar hydrogen and 200 RPM stirring speed. Typically, 40 mL $H_2O$, 0.4 g of T-MIL-88B-500, 5 mmol (312.5 mmol/$g_{cat}$.L) of HCOOH and 10 mmol (625 mmol/$g_{cat}$.L) of $CH_3I$ were added in Teflon-liner and reactor was sealed. After achieving the above-described conditions, 2 mL liquid sample was withdrawn from the reactor after regular intervals (1, 2, 4, 8, 12 and 24 h) for HPLC analysis. In the 2nd reaction system, aqueous phase $CO_2$ hydrogenation with $CH_3OH$ (10 mmol) and LiI (10 mmol) was performed over T-MIL-88B-500 (0.4 g) for 48 h at 150 °C, 40 mL $H_2O$, equimolar $H_2/CO_2$ under 70 bar at room temperature and 200 RPM stirring speed. After 48 h, the reactor was cooled to room temperature. Both liquid and gas samples were collected for product analysis, where gas sample was analysed through Shimadzu 2014 GC coupled with TCD and FID detectors, respectively.

### Computational methodology
The ReaxFF Fe/C/H/O force field employed in this work was originally developed to describe Fischer–Tropsch (FT) catalysis, and CO methanation and the hydrocarbon chain initiation[54,55]. The force field had been trained for hydrogen adsorption, dissociation and migration on iron and iron carbide surfaces, and binding energies of small hydrocarbon radicals and energy barriers for $CH_4$ dissociation on Fe(100) surface. To extend this force field to describe MIL-88B model, Fe-O-C valence angle parameters were re-optimized against Fe-complexes relevant to the MIL-88B metal cluster topology. Figure 10 shows the energetics of mono- and bi-dentate Fe(II)-complexes containing OH, $H_2O$, $CO_2$, $O_2$, acetate, peracetate and pyruvate ligands,

which may appear in the catalytic iron coordination complex. The reaction energies for the conversion of acetate into peracetate, and peracetate into pyruvate are reproduced properly in ReaxFF. In addition, the binding energies of $O_2$, $CO_2$ and pyruvate ligand (strongest) to Fe(II) metal are in good agreement with the DFT data. A full description of this force field is given in Supplementary Data 1.

To mimic the thermal transformation of MIL-88B, we performed reactive molecular dynamics simulations using ReaxFF force field parameters. First, the MIL-88B(Fe) is equilibrated at 300 K under *NPT* ensemble for 100 ps. We use the Nose-Hoover thermostat and Berendsen barostat to control the temperature and pressure, respectively. To thermally transform the MOF, we keep the MOF at either 1500 K (200 ps and 500 ps) or 2000 K (500 ps) under *NVT* ensemble. Such high temperatures are commonly used in reactive molecular dynamics simulations to accelerate the reactions and mimic the experimental timescale (h) within computational timescale (ns). To reach these high temperatures, the MOF is heated at a rate of 2 K/ps under *NPT* ensemble below 1500 K and under *NVT* ensemble at 1500 K and above. The *NVT* ensemble is used for temperatures ≥1500 K to avoid an extremely low-density structure due to high temperature, which is not the case experimentally. Molecular dynamics is simulated with timestep of 0.25 fs (<1500 K) or 0.1 fs (>1500 K) and the damping constant for thermostat and barostat are 100 fs and 1500 fs, respectively. After 500 ps at high temperatures (1500 K and 2000 K), we cool the MOF to 300 K with a cooling rate of 4 K/ps. During cooling, we simulate *NPT* ensemble for $T < 1500$ K and NVT ensemble for $T \geq 1500$ K.

## Data availability
All the data are available in the main text or the electronic supplementary information (ESI).

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

## Acknowledgements

The authors would like to thank the Faculty of Engineering, Monash University for financial support under the Researcher Accelerator Grant 2019. The Advanced Light Source synchrotron access was funded by Australian Research Council Discovery International Award (DP170104017). The authors would also acknowledge Monash Centre for Electron Microscopy (MCEM) for providing the microscopic analysis facilities. A.T. and A.S. received financial support from the Institute for Catalysis, Hokkaido University as part of their Strategic Research Fellowship grant scheme. This study was supported by the Cooperative Research Program of Institute for Catalysis, Hokkaido University (Proposal no. 19A1005). This research was also supported in part by the Monash eResearch Centre and eSolutions-Research Support Services through the use of the MonARCH HPC Cluster.

## Author contributions

Conceptualization: W.A., P.K. and A.T.; methodology: W.A., P.K., S.D., Y.K.S., A.S. and A.T.; investigation: W.A., P.K., R.L., S.D., Y.K.S., A.C.T.v.D., A.S. and A.T.; visualization: W.A., P.K., S.D., R.L. and A.T.; funding acquisition: A.S. and A.T.; project administration: A.T.; supervision: A.C.T.v.D., A.S. and A.T.; writing: W.A., P.K., R.L., S.D., Y.K.S., A.C.T.v.D. and AT. W.A. and P.K. have equal contributions.

## Competing interests

The authors declare no competing interests.
