## [Peer Review File · Nature Communications]

REVIEWER COMMENTS

Reviewer #1 (Remarks to the Author):

The authors prepared heterogeneous catalyst using MIL-88B, FeO and Fe₃O₄. The catalytic performance for methanol hydrocarboxylation with CO₂ and H₂ to produce acetic acid (AA) was studied. High AA yield and selectivity were achieved in the presence of CH₃I and LiI. I think the manuscript may become acceptable after revisions, but it should be re-reviewed. Please consider the following comments.

The introduction section

1. In the introduction, a recent published article should be cited and discussed (ACS Catal. 2021, 11, 8382–8398), which reported a HETEROGENEOUS Ni_xZn_yO catalyst that directly converted CO₂ to acetic acid and propionic acid with an overall selectivity of 77.1%. In this paper, high overall selectivity of acetic acid and propionic acid were achieved without additional substrates on a HETEROGENEOUS catalyst without a promoter. Obviously, it should be discussed in detail.
2. In the Introduction, the background of CO₂ utilization and acetic acid synthesis has not been sufficiently revealed, because the most important papers including Review papers are not cited.
3. The arrows to connect the reaction equation are inappropriate, do they indicate reversible reactions? Can the authors confirm that all the reactions or reaction steps are reversible? As far as I know, at least the methanol carbonylation (equation 1) is not reversible.
4. In the introduction, “This reaction system is highly complex due to the presence of multiple catalysts, stabilizing ligands and organic solvents. In many cases, the authors report a black precipitate, which is not explained but is likely to be the Ru or Rh catalyst, which demonstrates that the system is not stable”. As homogeneous catalysts, the selected catalytic system can even be recycled and reused, indicating they are stable. The discussion of the literature should be accurate.
5. In the introduction, “the most common industrial processes are carbonylation of methanol (MeOH) developed by BASF, Cativa and Monsanto”, Cativa is the name of the Iridium based methanol carbonylation technology, which was developed by BP.

The experimental

6. The experimental procedures for synthesis and characterization should be provided in a more detailed manner.

7. The impact of temperature has not been tested and discussed, as well as the ratio of CO₂ and H₂ pressure. The reaction time was very long compared to the amount of the reactant MeOH. The performance of the catalyst may be better at higher temperature.
8. I noticed that there was lack of the whole product distribution for every reaction referred in the manuscript. I think it is necessary to complete the product distribution of reactions and give the corresponding graphs of the product analysis. The percentage of the two acids in the total products is a key information to evaluate the performance of the catalysts. The selectivity data in the current manuscript should be mentioned with a clear definitive phrase, especially in abstract and the Conclusion section.
9. To highlight catalytic performance towards acetic acid in CO₂ hydrogenation in this manuscript, the comparison of reported results, especially heterogeneous catalysts, should be given.
10. The GC graphs of the gaseous sample (Fig. S4a and S4c) are not correct, because the FID detector cannot get the signal of CO₂.
11. It is known that CH₃I does not dissolve in water. What is the role of water? D₂O labelling test is needed to make this more clear. What will happen without the water as solvent? How about the results in organic solvents? These should be supplied in SI.
12. The amounts of CH₃I, methanol, LiI, and the catalyst dosage should be given in the legends of Figures 6-8.
13. In Figure 7, the change of the amount of CH₃I, methanol, and LiI with time should be given.
14. The reaction using ¹³C labelled MeOH as substrate should be done to confirm its participation in the reaction and its position in the acetic acid product.
15. The authors gives three Fe-related materials with different composition in the manuscript and deduce that the Fe₀ and Fe₃O₄ should be the key component for the high AA yield and selectivity by the XRD and XPS characterizations as well as reference. However, the discussion is not convinced because the characterization details were missing. The active sites in the manuscript are still skeptical based on existed information. For example, I doubt that Fe species would be oxidized without protection and the following characterization could be inaccurate. And 1-3 heterogeneous catalysts with other metal (not Fe) should be applied in the condition of the manuscript to detect the active sites. The in-situ characterizations (for example, in situ-XPS or in situ EXAFS) would give the useful information of the reaction mechanism or guide to the final conclusion.
16. The authors speculate that the acetic acid formation passes through the formic acid route and give the possible mechanism in Figure 10. To confirm this mechanism, only FA experiment (Figure 9) is far from sufficient. For example, the direct insertion of CO₂ is also possible in the existence of MeOH and LiI based on existed information. Moreover, reaction of CO₂ and H₂ to form HCOOH is a well known reversible reaction. Systematic control experiments including in situ FTIR are required to confirm this reaction route.

17. I suggests that the author also supplement control experiments to test or exclude other possible intermediates (such as methyl formate, CO, dimethyl ether). The performance of T-MIL-88B in CO₂ hydrogenation without CH₃I or MeOH should be provided.

18. The mechanism in Figure 10 still needs to answer the roles of the Fe₃O₄ and the support. It still needs more evidence or reference to support the reaction steps. In addition, the equation around (1) in Figure 10 is incorrect, both material balance and the arrow direction.

19. In Section 3.2.4, “followed by formation of CH₃Rh*^I due to the insertion of CH₃I into a Rh* complexing catalyst 13”, the insertion should be “oxidative addition”.

20. In general, the logic of data organization and discussion of this manuscript are not clear enough.

21. There are some writing mistakes, for example, in line 376 of page 18, it should be “after 48h of reaction” instead of “after 48 of reaction”.

Reviewer #2 (Remarks to the Author):

The article by Tanksale and a group of authors is focused on the catalytic conversion of CO₂ to acetic acid in an aqueous medium, using thermally "transformed" MIL-MOFs (MIL-88B and MIL-100) as catalysts. I believe this is a significant field of study, and the shown catalytic results are promising, but several points need to be taken into consideration and altered before a positive evaluation can be given. I will list the most critical issues below:

There is a lack of analytic methods that would support some of the authors' claims in the discussion and conclusion section.

For example, there is a lack of PXRD data for the raw MIL-88B material the authors prepared as a starting material for their T-MIL-88B catalyst. The only XRD presented for these materials is fig 2c in the manuscript that shows thermally "transformed" MIL-88B material, showing more or less amorphous matrix from which the authors identified the peaks for Fe₃O₄ and Fe(0). It is not clear why the authors chose to use the MIL-88B synthesis from 2017 and not the original one. Is the MIL-88B the starting compound for their calcination?

All the analytical data that could help follow and potentially repeat these experiments must become a part of this manuscript. Also, please show (and refer to) the calculated XRD data for the products and reactants in the discussion.

The authors claim iron(0) in their T-MIL-88 catalyst, which is not present in their calcinated MIL-101 catalyst. The presence of these iron nanoparticles is further taken as a critical factor in the superior performance of the calcinated MIL-88B catalyst. The presence of Fe(0) and the simulation from which the ratio was calculated does not look absolutely convincing in the fig 4a. The caption or the text discussing the XPS data presented in figure 4 should be a bit more informative to explain the basis for the presented results. However, in my opinion, the authors should support their claims with a more precise method for the determination of iron species than the XPS and PXRD, best Mossbauer spectroscopy. I find it hard to imagine why the iron oxide nanoparticles in MIL-88 would be reduced, and those in MIL-101, available from the very initiation of thermal treatment in the reductive atmosphere, would remain in their starting form.

I see no discussion or comment on the potential role of sodium in catalysis, while at the same time, the calcinated MIL-88B contains almost 20% of sodium. Also, why is such a large quantity of sodium present in the MIL-88B sample after several washing cycles?

The discussion on the thermal degradation of MIL-88B is highly speculative. While the discussion is based on the published computational study of thermal degradation of MOF (ref 24 in the manuscript), both the metal node and the linker differ significantly from the ones presented here and cannot be used to corroborate the proposed degradation mechanism in iron MOF. Zirconium carboxylate MOFs are a class for themselves for many well-known reasons. The discussion on thermal degradation of Fe-MIL-88B in the presented study has no experimental or computational proof, making Figure 2 misleading. It represents this speculative process, and the final figure, figure 2e, is likely derived from their ref 24 data. It shows biphenyl ligands in the product matrix, and a large amount of phenyl species here is also not in line with the elemental analysis results presented in the SI. Again, what is happening with the 20% of sodium in this process?

Why not add LiOH in the synthesis of MIL-88B in the first place to serve as co-catalyst?

Is there any data on thermal behavior and weight-loss steps in the experiment that would correspond to this thermal transformation? At least, what is the weight of the calcinated product when the thermal treatment is finished? Could the authors perform a similar experiment in TGA? Or at least the experiment in Ar, where the MIL-88B would be kept at 500 deg C for 5 hours, in line with the calcination of MIL-88B proposed in the manuscript. The DSC or DTA data could also help in discussing the thermal events during the calcination.

Have the authors assessed the catalytic activity on their nascent MOF materials and compared their activity to thermally-transformed samples to show the advantages of calcination approach, or does such data already exist for this particular class of MOFs and these target reactions?

I did not focus on the Fe/CBEA catalyst due to its exclusion from the catalytic assessment, but what would be a conclusion about this material? Why would the iron egress from it while remaining intact in Fe@MIL-101 case?

If I understood correctly, the authors did not observe iron carbonates after the catalytic cycle? Could the authors comment on this? The binding of the carbon dioxide is the key step for catalysis as can be seen from the proposed mechanism, and CO₂ is in large excess, and it also dissolves better in water than hydrogen. If the PXRD is the main technique for determining the presence of iron carbonate, it should be noted that this compound can (and often is) amorphous.

Reviewer #3 (Remarks to the Author):

This work entitled "Aqueous phase conversion of CO₂ into acetic acid over thermally transformed MIL-88B" examine the series of catalysts for acetic acid production. The authors showed that the thermally transformed T-MIL-88B has a high catalytic activity and stability for aqueous phase CO₂ transformation into acetic acid. The catalytic activity and the structural property of T-MIL-88B were compared with Fe/CBEA and thermally transformed Fe deposited on MIL-101 (T-Fe/MIL-101). The authors reported that T-MIL-88B consisted of both Fe⁰ and Fe₃O₄ phases making superior activity for acetic acid production than the other catalysts. The subject is definitely of particular importance both from the fundamental and practical points of view towards the rational design of cost-efficient catalysts. I consider, however, that various issues, listed below, hinder the publication of the article in the present form. Please address the following questions.

The details revision are as follows

1. Although the authors mentioned that Fe⁰ and Fe₃O₄ phases of T-MIL-88B catalyst are responsible for the higher activity, evidence is missing. Authors should show the active site responsible for the higher activity.
2. According to the Page 5, Line no. 146, the authors stated the XPS C 1s calibrated to 284.8 eV, however, Fig. 4 C 1s C-C peak shows different binding energy. Please do the charge correction once again.
3. XPS peak fitting of Fe 2p peak should be modified.

4. Although activity results are interesting, more shreds of evidence are required to find out the active site of the catalysts.

5. Please calculate the Fe dispersión on various supports.

6. Please include the CO₂-Temperature programmed desorption studies of all catalysts.

REVIEWER COMMENTS

Reviewer #1 (Remarks to the Author):

The authors prepared heterogeneous catalyst using MIL-88B, FeO and Fe₃O₄. The catalytic performance for methanol hydrocarboxylation with CO₂ and H₂ to produce acetic acid (AA) was studied. High AA yield and selectivity were achieved in the presence of CH₃I and LiI. I think the manuscript may become acceptable after revisions, but it should be re-reviewed. Please consider the following comments.

The introduction section

1. In the introduction, a recent published article should be cited and discussed (ACS Catal. 2021, 11, 8382–8398), which reported a HETEROGENEOUS Ni_xZn_yO catalyst that directly converted CO₂ to acetic acid and propionic acid with an overall selectivity of 77.1%. In this paper, high overall selectivity of acetic acid and propionic acid were achieved without additional substrates on a HETEROGENEOUS catalyst without a promoter. Obviously, it should be discussed in detail.

Response 1: We discussed the suggested article in the introduction section of revised manuscript.

2. In the Introduction, the background of CO₂ utilization and acetic acid synthesis has not been sufficiently revealed, because the most important papers including Review papers are not cited.

Response 2: We have extended the background of CO₂ utilization and acetic acid synthesis as well as cited the relevant review articles in the revised manuscript. We also included the most recent articles of AA synthesis via CO₂ feedstock in the introduction section.

3. The arrows to connect the reaction equation are inappropriate, do they indicate reversible reactions? Can the authors confirm that all the reactions or reaction steps are reversible? As far as I know, at least the methanol carbonylation (equation 1) is not reversible.

Response 3: Thank you so much for this comment. We revised equations 1, 3, 4 and 5 in the manuscript as described in references 15, 16 and 25 of the revised manuscript.

15. Rohmann K, Kothe J, Haenel MW, Englert U, Hölscher M, Leitner W. Hydrogenation of CO₂ to formic acid with a highly active ruthenium acridophos complex in DMSO and DMSO/water. *Angewandte Chemie International Edition* **55**, 8966-8969 (2016).
16. Qian Q, Zhang J, Cui M, Han B. Synthesis of acetic acid via methanol hydrocarboxylation with CO₂ and H₂. *Nature communications* **7**, 1-7 (2016).
25. Budiman AW, *et al.* Review of acetic acid synthesis from various feedstocks through different catalytic processes. *Catalysis Surveys from Asia* **20**, 173-193 (2016).

4. In the introduction, "This reaction system is highly complex due to the presence of multiple catalysts, stabilizing ligands and organic solvents. In many cases, the authors report a black precipitate, which is not explained but is likely to be the Ru or Rh catalyst, which demonstrates that the system is not stable". As homogeneous catalysts, the selected catalytic system can even be recycled and reused, indicating they are stable. The discussion of the literature should be accurate.

Response 4: We included the statement about the stable catalytic activity of Rh and Ru based homogeneous catalysts in the revised manuscript as described below:

"This reaction system is highly complex due to the presence of multiple catalysts, stabilizing ligands and organic solvents. In many cases, the authors report a black precipitate, which is not explained but is likely to be the Ru or Rh catalyst, which demonstrates that the system is not stable in these reaction conditions. However, authors also demonstrated the stable catalytic activity of Rh and Ru based homogeneous catalysts for five cycle during AA synthesis via methanol hydrocarboxylation reaction by using Imidazole ligand and LiI promoter in DMI solvent¹⁶."

5. In the introduction, "the most common industrial processes are carbonylation of methanol (MeOH) developed by BASF, Cativa and Monsanto", Cativa is the name of the Iridium based methanol carbonylation technology, which was developed by BP.

Response 5: We have changed the statement in the revised manuscript as follow:
“Among various chemical routes, the most common industrial AA synthesis method is carbonylation of methanol (MeOH), where, AA is produced through different processes such as BASF, Cativa and Monsanto in the presence of homogeneous Cobalt, Iridium and Rhodium catalysts, respectively.”

The experimental

6. The experimental procedures for synthesis and characterization should be provided in a more detailed manner.

Response 6: We have modified synthesis and characterization section of revised manuscript.

7. The impact of temperature has not been tested and discussed, as well as the ratio of CO₂ and H₂ pressure. The reaction time was very long compared to the amount of the reactant MeOH. The performance of the catalyst may be better at higher temperature.

Response 7: The maximum operating pressure of our autoclave batch reactor was 100 bar. We performed reaction at 70 bar at room temperature which was reached above 95 bar at 150 °C. Therefore, it did not allow us to conduct the reaction at further higher temperature. Moreover, maximum operating temperature of Teflon liner was 150 °C which was another limitation to run the experiment at this maximum temperature.

8. I noticed that there was lack of the whole product distribution for every reaction referred in the manuscript. I think it is necessary to complete the product distribution of reactions and give the corresponding graphs of the product analysis. The percentage of the two acids in the total products is a key information to evaluate the performance of the catalysts. The selectivity data in the current manuscript should be mentioned with a clear definitive phrase, especially in abstract and the Conclusion section.

Response 8: Only two products were detected in liquid samples which have been reported in the manuscript.

9. To highlight catalytic performance towards acetic acid in CO₂ hydrogenation in this

manuscript, the comparison of reported results, especially heterogeneous catalysts, should be given.

Response 9: We followed the instruction of respected reviewer and included the comparison statement of present study with most recent published article of acetic acid production from aqueous phase CO₂ hydrogenation in revised manuscript as described below.

“Recently, Wang et al. reported the acetate production through formate (HCOO⁻) and carbene (*CH₂) intermediate reaction pathway using hexagonal closed packed cobalt (HCP-Co) catalyst and NaOH additive which provided maximum 9.5% acetate yield after 6 h of reaction at 300 °C, 0.5 M NaOH, 40% water filling, and 40 mmol Co under 15 bar CO₂ and 35 bar H₂ atmosphere. However, the catalyst was inactive at lower temperature (< 200 °C) ³⁰. In comparison, the present study envisioned the 504 mmol/g_{cat}.L of AA yield with 92.4% AA selectivity after 21 h of reaction over T-MIL-88B at 150 °C, H₂/CO₂= 1, CH₃I= 10 mmol, catalyst amount= 400 mg, H₂O= 40 mL and stirring speed= 200 RPM.”

10. The GC graphs of the gaseous sample (Fig. S4a and S4c) are not correct, because the FID detector cannot get the signal of CO₂.

Response 10: The GC was equipped with a methanizer before the FID detector. Therefore, GC-FID detector can get the signals of CO₂ and CO. This is a common setup to increase the sensitivity of CO₂ and CO detection in the GC.

11. It is known that CH₃I does not dissolve in water. What is the role of water? D₂O labelling test is needed to make this more clear. What will happen without the water as solvent? How about the results in organic solvents? These should be supplied in SI.

Response 11: The CH₃I solubility in water is 14 g/L at 20 °C. The solubility of CH₃I in organic solvents is higher as compared to water. But, we performed this reaction in water only because the process for separation of acetic acid from water is well established on industrial scale.

12. The amounts of CH₃I, methanol, LiI, and the catalyst dosage should be given in the legends of Figures 6-8.

Response 12: We followed the instruction of respected reviewer and added the amounts of CH₃I, methanol, LiI, and the catalyst in the legends of revised Figures 7-10.

13. In Figure 7, the change of the amount of CH₃I, methanol, and LiI with time should be given.

Response 13: CH₃I and LiI were not detected in HPLC analysis.

14. The reaction using ¹³C labelled MeOH as substrate should be done to confirm its participation in the reaction and its position in the acetic acid product.

Response 14: Literature already reported the ¹³C labelling experiment for ¹³CH₃COOH synthesis via CO₂, H₂ and ¹³CH₃I in water solvent using Rh-based homogeneous catalyst. We have cited this article in proposed reaction pathway section of the revised manuscript as mentioned below:

“Yatabe et al. described AA synthesis via CO₂ hydrogenation with CH₃I additive in aqueous phase using water-soluble Rh-based homogeneous catalyst. In reported study, ¹³CH₃I isotopic labelling experiment confirmed the presence of ¹³CH₃COOH, where, ¹³CH₃ group origin was linked with ¹³CH₃³¹.”

15. The authors gives three Fe-related materials with different composition in the manuscript and deduce that the FeO and Fe₃O₄ should be the key component for the high AA yield and selectivity by the XRD and XPS characterizations as well as reference. However, the discussion is not convinced because the characterization details were missing. The active sites in the manuscript are still skeptical based on existed information. For example, I doubt that Fe species would be oxidized without protection and the following characterization could be inaccurate. And 1-3 heterogeneous catalysts with other metal (not Fe) should be applied in the condition of the manuscript to detect the active sites. The in-situ characterizations (for

example, in situ-XPS or in situ EXAFS) would give the useful information of the reaction mechanism or guide to the final conclusion.

Response 15: Authors would like to thank their respected reviewer for his valuable comment. Fe particles in T-MIL-88B are mostly encapsulated in carbon matrix which was most likely the reason of maintaining its oxidation states after reaction as shown in XRD patterns of spent T-MIL-88B (Figure 1.c). However, the synthesis of other metal-based MOFs and their thermal transformations are out of scope of the present study. But, we will consider this useful suggestion in our future work.

16. The authors speculate that the acetic acid formation passes through the formic acid route and give the possible mechanism in Figure 10. To confirm this mechanism, only FA experiment (Figure 9) is far from sufficient. For example, the direct insertion of CO₂ is also possible in the existence of MeOH and LiI based on existed information. Moreover, reaction of CO₂ and H₂ to form HCOOH is a well known reversible reaction. Systematic control experiments including in situ FTIR are required to confirm this reaction route.

Response 16: It would be exceedingly difficult to conduct *in situ* FTIR in liquid phase catalytic reaction at high pressure and temperature. Currently there is no setup we are aware of which can conduct this experiment. Due to solvent the signal to noise ratio will be very weak.

17. I suggests that the author also supplement control experiments to test or exclude other possible intermediates (such as methyl formate, CO, dimethyl ether). The performance of T-MIL-88B in CO₂ hydrogenation without CH₃I or MeOH should be provided.

Response 17: In the absence of CH₃I or MeOH we did not detect any acetic acid. Only formic acid is formed in this case.

18. The mechanism in Figure 10 still needs to answer the roles of the Fe₃O₄ and the support. It still needs more evidence or reference to support the reaction steps. In addition, the equation around (1) in Figure 10 is incorrect, both material balance and the arrow direction.

Response 18: We proposed the reaction mechanism on the basis of present results over studied thermally transformed MIL-88B. The literature references were also cited to support this reaction pathway. We have checked the equation around 1 in Figure 11 of the revised manuscript. We found it correct and it represented the reaction of LiI with methanol as we already mentioned in introduction section equation 3 of the revised manuscript.

19. In Section 3.2.4, “followed by formation of CH₃Rh*I due to the insertion of CH₃I into a Rh* complexing catalyst 13”, the insertion should be “oxidative addition”.

Response 19: We modified the statement as suggested.

20. In general, the logic of data organization and discussion of this manuscript are not clear enough.

Response 20: To avoid confusions, we tried to clarify the statements by adding more discussion as well as we compared the present results with published study in the revised manuscript.

21. There are some writing mistakes, for example, in line 376 of page 18, it should be “after 48h of reaction” instead of “after 48 of reaction”.

Response 21: We have revised the manuscript to address the typographical errors.

Reviewer #2 (Remarks to the Author):

The article by Tanksale and a group of authors is focused on the catalytic conversion of CO₂ to acetic acid in an aqueous medium, using thermally "transformed" MIL-MOFs (MIL-88B and MIL-100) as catalysts. I believe this is a significant field of study, and the shown catalytic results are promising, but several points need to be taken into consideration and altered before a positive evaluation can be given. I will list the most critical issues below:

Comment 1: There is a lack of analytic methods that would support some of the authors' claims in the discussion and conclusion section. For example, there is a lack of PXRD data for the raw MIL-88B material the authors prepared as a starting material for their T-MIL-88B catalyst. The only XRD presented for these materials is fig 2c in the manuscript that shows thermally "transformed" MIL-88B material, showing more or less amorphous matrix from which the authors identified the peaks for Fe₃O₄ and Fe(0). It is not clear why the authors chose to use the MIL-88B synthesis from 2017 and not the original one. Is the MIL-88B the starting compound for their calcination? All the analytical data that could help follow and potentially repeat these experiments must become a part of this manuscript. Also, please show (and refer to) the calculated XRD data for the products and reactants in the discussion.

Response 1: We firstly synthesized MIL-88B and performed the thermal transformation of this material. The PXRD pattern of pristine MIL-88B was added in the revised manuscript ESI (electronic supplementary information) as Figure S.1.b.

Comment 2: The authors claim iron(0) in their T-MIL-88 catalyst, which is not present in their calcinated MIL-101 catalyst. The presence of these iron nanoparticles is further taken as a critical factor in the superior performance of the calcinated MIL-88B catalyst. The presence of Fe(0) and the simulation from which the ratio was calculated does not look absolutely convincing in the fig 4a. The caption or the text discussing the XPS data presented in figure 4 should be a bit more informative to explain the basis for the presented results. However, in my opinion, the authors should support their claims with a more precise method for the determination of iron species than the XPS and PXRD, best Mossbauer spectroscopy. I find it hard to imagine why the iron oxide nanoparticles in MIL-88 would be reduced, and those in

MIL-101, available from the very initiation of thermal treatment in the reductive atmosphere, would remain in their starting form.

Response 2: We repeated the XPS characterization of T-MIL-88B as well as conducted this analysis for MIL-88B. We also included additional discussion in the XPS section for better understanding.

In MIL-88B, iron was present as Fe_3O_4 in the framework of this MOF (Figure 4.a). In Fe/MIL-101, Fe was impregnated over MIL-101 and XPS results suggested the absence of Fe_3O_4 (Figure 4.a). However, PXRD analysis demonstrated the likelihood of $\alpha\text{-Fe}_2\text{O}_3$ phases (Figure S.1.a) in this catalyst. Therefore, it might be possible that thermal transformation behaviour of MIL-88B was different as compared to Fe/MIL-101 at identical temperature (500 °C).

Comment 3: I see no discussion or comment on the potential role of sodium in catalysis, while at the same time, the calcinated MIL-88B contains almost 20% of sodium. Also, why is such a large quantity of sodium present in the MIL-88B sample after several washing cycles?

Response 3: We thank reviewer for this good observation. Sodium present in the solid phase is likely to behave as a spectator ion in the liquid phase (water) reaction conditions and may not affect the acidity/basicity of the liquid phase. If sodium is to affect the reaction mechanism by neutralising formic intermediate or acetic acid by formation of formate or acetate salts, the same catalyst activity could not have been observed in multiple recycle reaction cycles.

Comment 4: The discussion on the thermal degradation of MIL-88B is highly speculative. While the discussion is based on the published computational study of thermal degradation of MOF (ref 24 in the manuscript), both the metal node and the linker differ significantly from the ones presented here and cannot be used to corroborate the proposed degradation mechanism in iron MOF. Zirconium carboxylate MOFs are a class for themselves for many well-known reasons. The discussion on thermal degradation of Fe-MIL-88B in the presented study has no experimental or computational proof, making Figure 2 misleading. It represents this speculative process, and the final figure, figure 2e, is likely derived from their ref 24 data.

It shows biphenyl ligands in the product matrix, and a large amount of phenyl species here is also not in line with the elemental analysis results presented in the SI. Again, what is happening with the 20% of sodium in this process?

Response 4: A detailed ReaxFF reactive molecular dynamics simulation was conducted and a new section on this analysis is presented in the revised manuscript which support the initial findings.

Comment 5: Why not add LiOH in the synthesis of MIL-88B in the first place to serve as co-catalyst?

Response 5: LiI was a co-catalyst instead of LiOH. It is the iodide ion which results in C-C via the formation of methyl iodide as an intermediate.

Comment 6: Is there any data on thermal behavior and weight-loss steps in the experiment that would correspond to this thermal transformation? At least, what is the weight of the calcinated product when the thermal treatment is finished? Could the authors perform a similar experiment in TGA? Or at least the experiment in Ar, where the MIL-88B would be kept at 500 deg C for 5 hours, in line with the calcination of MIL-88B proposed in the manuscript. The DSC or DTA data could also help in discussing the thermal events during the calcination.

Response 6: We conducted TGA analysis of MIL-88B in 5% H_2/N_2 gas mixture, and N_2 gas, respectively. The weight loss and derivative weight loss profiles of these analysis were added as Figure 5.b. The detailed discussions were also included in the revised manuscript.

Comment 7: Have the authors assessed the catalytic activity on their nascent MOF materials and compared their activity to thermally-transformed samples to show the advantages of calcination approach, or does such data already exist for this particular class of MOFs and these target reactions?

Response 7: According to our best knowledge, we did not find the information about CO₂ hydrogenation to acetic acid using these MOFs in literature. We performed this reaction with thermally transformed MOFs because we tried to compare the activity of reduced iron based studied catalysts (Fe/CBEA, Fe/MIL-101 and MIL-88B).

Comment 8: I did not focus on the Fe/CBEA catalyst due to its exclusion from the catalytic assessment, but what would be a conclusion about this material? Why would the iron egress from it while remaining intact in Fe@MIL-101 case?

Response 8: In reduced Fe/CBEA, iron particles might be present over the surface of catalyst support. Therefore, it was most likely possible that iron was leached out during the aqueous phase CO₂ hydrogenation as explained in PXRD section. In T-Fe/MIL-101, iron particles were mainly encapsulated in carbon matrix after thermal transformation under reduction atmosphere which was most likely the reason of stability of this catalyst after reaction as shown in PXRD patterns (Figure 1.b).

Comment 9: If I understood correctly, the authors did not observe iron carbonates after the catalytic cycle? Could the authors comment on this? The binding of the carbon dioxide is the key step for catalysis as can be seen from the proposed mechanism, and CO₂ is in large excess, and it also dissolves better in water than hydrogen. If the PXRD is the main technique for determining the presence of iron carbonate, it should be noted that this compound can (and often is) amorphous.

Response 9: Literature reported the XRD peaks of ferrous carbonate of spent Fe nanoparticles after FA and AA production through hydrothermal CO₂ reduction using Fe nanoparticles as stoichiometric reagent. We have described and cited this published study in the introduction section as:

“He et. al. report FA and AA production via hydrothermal CO₂ reduction with Fe nanoparticles as stoichiometric reagent in which they are converted into ferrous carbonate²⁹.”

However, we did not observe the ferrous carbonate peaks for PXRD of spent catalysts.

Reviewer #3 (Remarks to the Author):

This work entitled "Aqueous phase conversion of CO₂ into acetic acid over thermally transformed MIL-88B" examine the series of catalysts for acetic acid production. The authors showed that the thermally transformed T-MIL-88B has a high catalytic activity and stability for aqueous phase CO₂ transformation into acetic acid. The catalytic activity and the structural property of T-MIL-88B were compared with Fe/CBEA and thermally transformed Fe deposited on MIL-101 (T-Fe/MIL-101). The authors reported that T-MIL-88B consisted of both Fe⁰ and Fe₃O₄ phases making superior activity for acetic acid production than the other catalysts. The subject is definitely of particular importance both from the fundamental and practical points of view towards the rational design of cost-efficient catalysts. I consider, however, that various issues, listed below, hinder the publication of the article in the present form. Please address the following questions.

The details revision are as follows:

1. Although the authors mentioned that Fe⁰ and Fe₃O₄ phases of T-MIL-88B catalyst are responsible for the higher activity, evidence is missing. Authors should show the active site responsible for the higher activity.

Response 1: We further thermally transformed MIL-88B catalyst at different temperatures (495 °C, 500 °C and 505 °C) and compared the PXRD patterns of these catalysts (T-MIL-88B-495, T-MIL-88B-500 and T-MIL-88B-505) as shown Figure S1.c of ESI (electronic supplementary information). The PXRD characteristic peaks clearly confirms the presence of different Phases of Fe₂O₃ and Fe⁰ in T-MIL-88B. While, diffraction peaks of T-MIL-88B-495 demonstrated the mainly Fe₃O₄ phases and T-MIL-88B-505 presented the major peaks of Fe₃C and Fe⁰, and few low intensity peaks of Fe₃O₄ phases. We also conducted TGA analysis (Figure 5.b) in 5%H₂/N₂ gas mixture and N₂ gas. A new additional derivative weight loss peak was emerged between 495-530 °C under 5%H₂/N₂ mixture which might be related with further transformation of MIL-88B into Fe⁰-carbon and Fe₃C-carbon composites. Therefore, PXRD and TGA results suggested the presence of Fe₃O₄ and Fe⁰ which were most likely responsible for

high activity of T-MIL-88B. The results and discussions of these characterization were included in the revised manuscript and electronic supplementary information.

2. According to the Page 5, Line no. 146, the authors stated the XPS C 1s calibrated to 284.8 eV, however, Fig. 4 C 1s C-C peak shows different binding energy. Please do the charge correction once again.

Response 2: As per the reviewer's suggestion, the charge correction has been performed in C 1s XPS spectra (Figure 3.c).

3. XPS peak fitting of Fe 2p peak should be modified.

Response 3: XPS peak fitting of iron for all catalysts has been modified. We also repeated XPS of T-MIL-88B. In addition, the MIL-88B XPS analysis was included for better understanding.

4. Although activity results are interesting, more shreds of evidence are required to find out the active site of the catalysts.

Response 4: Please check response 1, 5 and 6 for additional characterization (PXRD, TGA, CO pulse chemisorption for Fe metal dispersion and CO₂-TPD) of T-MIL-88B.

5. Please calculate the Fe dispersión on various supports.

Response 5: We conducted CO pulse chemisorption of best performance catalyst and Fe metal dispersion of this catalyst was added in the revised manuscript.

6. Please include the CO₂-Temperature programmed desorption studies of all catalysts.

Response 6: We performed CO₂-TPD analysis of best catalyst (T-MIL-88B) and included it in the revised manuscript as Figure 6.

REVIEWERS' COMMENTS

Reviewer #1 (Remarks to the Author):

I think the revised manuscript is acceptable.

Reviewer #2 (Remarks to the Author):

The authors extensively reviewed their manuscript, and most of my previous comments were answered satisfactorily. Only a few minor issues here:

Comment to Response 1, rev2: The diffractograms are still not presented in a satisfactory way. The starting angle is too high. Please show the patterns from at least 5 degrees 2Theta (preferably 3 degrees), and include the simulated patterns from the single-crystal data in both figures.

Please also include simulated patterns for iron oxides and elemental iron observed in the samples. Some Bragg reflections in the presented XRD data, assigned to iron oxides, do not seem to fit.

Comment to Response 2: This material displays interesting thermal behavior in the reductive atmosphere. Is the observed formation of Fe(0) and Fe₃C phase in the 495-505 °C range reproducible? Have the authors checked which solid phases are present in the products heated above 530 °C?

Comment to Response 3: My question was how the sodium survived the extensive washing (DMF, methanol). In which form is sodium present here? The quantity of sodium in the sample is similar to iron. How are no signs of any solid sodium phase observable in diffraction studies? Most sodium inorganic phases are crystalline.

Have the authors also considered using high-resolution TEM-EDAX to experimentally establish the distribution of these two metals in their samples? It may further corroborate their conclusions about the degradation mechanism and help establish the size of the iron-containing particles in the carbon matrix.

Response to Reviewer #2:

The authors extensively reviewed their manuscript, and most of my previous comments were answered satisfactorily. Only a few minor issues here:

Comment to Response 1, rev2: The diffractograms are still not presented in a satisfactory way. The starting angle is too high. Please show the patterns from at least 5 degrees 2Theta (preferably 3 degrees), and include the simulated patterns from the single-crystal data in both figures. Please also include simulated patterns for iron oxides and elemental iron observed in the samples. Some Bragg reflections in the presented XRD data, assigned to iron oxides, do not seem to fit.

All the XRD plots already start at $2\theta = 5^\circ$. The low angle XRD ($2 - 5^\circ$) is useful for ordered mesoporous structures, however, after thermal treatment of MIL-88B, our catalyst does not exhibit ordered structure as is evident by the broad carbon peak between $5 - 30^\circ$.

Comment to Response 2: This material displays interesting thermal behavior in the reductive atmosphere. Is the observed formation of Fe(0) and Fe₃C phase in the 495-505 °C range reproducible? Have the authors checked which solid phases are present in the products heated above 530 °C?

Firstly, we did not observe Fe₃C phase in our catalyst. The decomposition behaviour is reproducible in a precisely controlled temperature environment. Since small changes in temperature can result in phase change, due care was taken to measure the catalyst temperature during thermal treatment. We did not study higher temperatures for thermal treatment since further reduction at higher temperatures would result in poor catalyst performance.

Comment to Response 3: My question was how the sodium survived the extensive washing (DMF, methanol). In which form is sodium present here? The quantity of sodium in the sample is similar to iron. How are no signs of any solid sodium phase observable in diffraction studies? Most sodium inorganic phases are crystalline.

Have the authors also considered using high-resolution TEM-EDAX to experimentally establish the distribution of these two metals in their samples? It may further corroborate their conclusions about the degradation mechanism and help establish the size of the iron-containing particles in the carbon matrix.

We agree with the reviewer that the effect of sodium may be explored in greater detail, and a more rigorous washing protocol may reduce the amount of leftover sodium. However, the leftover salt is common in MOF synthesis due to the solvent trapped in the pores. This salt concentration is expected to increase in the thermally transformed MOF as the solvent evaporates and the organic linkers break down, especially once the MOF has undergone a mass loss greater than 50% as is the case here. We do not expect the leftover sodium in crystalline form; as we expect sodium to remain dispersed in the carbonaceous matrix. There is no evidence of crystals in the XRD patterns of several prepared catalysts. Since sodium is unlikely to play any role in the catalysis of the reaction, we keep the present manuscript limited to the synergistic role of Fe/Fe(III)-oxides for CO₂ conversion.

However, to include the reviewer's valid observation, we have added the following line in the manuscript for future investigations.

"The presence of sodium and other non-volatile impurities increases substantially in a thermally transformed MOF. The effect of such impurities on catalyst morphology may be explored in expanding research."